# Collaborative Memory: Multi-User Memory Sharing in LLM Agents with Dynamic Access Control

## Abstract

Complex tasks are increasingly delegated to ensembles of specialized LLM-based agents that reason, communicate, and coordinate actions—both among themselves and through interactions with external tools, APIs, and databases. While persistent memory has been shown to enhance single-agent performance, most approaches assume a monolithic, single-user context—overlooking the benefits and challenges of knowledge transfer across users under dynamic, asymmetric permissions. We introduce **Collaborative Memory**, a framework for multi-user, multi-agent environments with asymmetric, time-evolving access controls encoded as bipartite graphs linking users, agents, and resources. Our system maintains two memory tiers: (1) *private memory*—private fragments visible only to their originating user; and (2) *shared memory*—selectively shared fragments. Each fragment carries immutable provenance attributes (contributing agents, accessed resources, and timestamps) to support retrospective permission checks. Granular *read policies* enforce current user–agent–resource constraints and project existing memory fragments into filtered transformed views. *Write policies* determine fragment retention and sharing, applying context-aware transformations to update the memory. Both policies may be designed conditioned on system, agent, and user-level information. Our framework enables safe, efficient, and interpretable cross-user knowledge sharing, with provable adherence to asymmetric, time-varying policies and full auditability of memory operations.

## 1 Introduction

The theory of Distributed Cognition Hutchins (1995) posits that cognitive processes are not confined to individual minds but are distributed across groups of people, artifacts, and their interactions within an environment. From this perspective, a collective system—comprising individuals and shared external representations—can function as a unified cognitive entity, capable of collaborative reasoning and complex problem-solving. This conceptual framework aligns closely with recent advances in multi-agent systems Guo et al. (2024); Tran et al. (2025a), where ensembles of specialized agents—often instantiated as large language models (LLMs)—are orchestrated to solve problems that exceed the capacity of any single model. These agents not only communicate and coordinate with one another but also interface with external resources such as tools, APIs, and structured data sources Yao et al. (2022); Schick et al. (2023).

A key enabler of such collaboration is persistent memory. Recent work has demonstrated that equipping LLM agents with long-term memory significantly enhances their ability to reason over extended horizons Lewis et al. (2020); Melz (2023). For example, MemGPT introduces an operating system-inspired abstraction for managing contextual memory over time Packer et al. (2023); MemTree employs a hierarchical memory structure for organizing and retrieving information dynamically Rezazadeh et al. (2024); and GraphRAG constructs a graph-based memory of entities and relations to enable structured knowledge access Edge et al. (2024); Shinn et al. (2023). These systems allow agents to store, recall, and build upon prior experiences in a manner that supports more coherent and context-aware behavior.

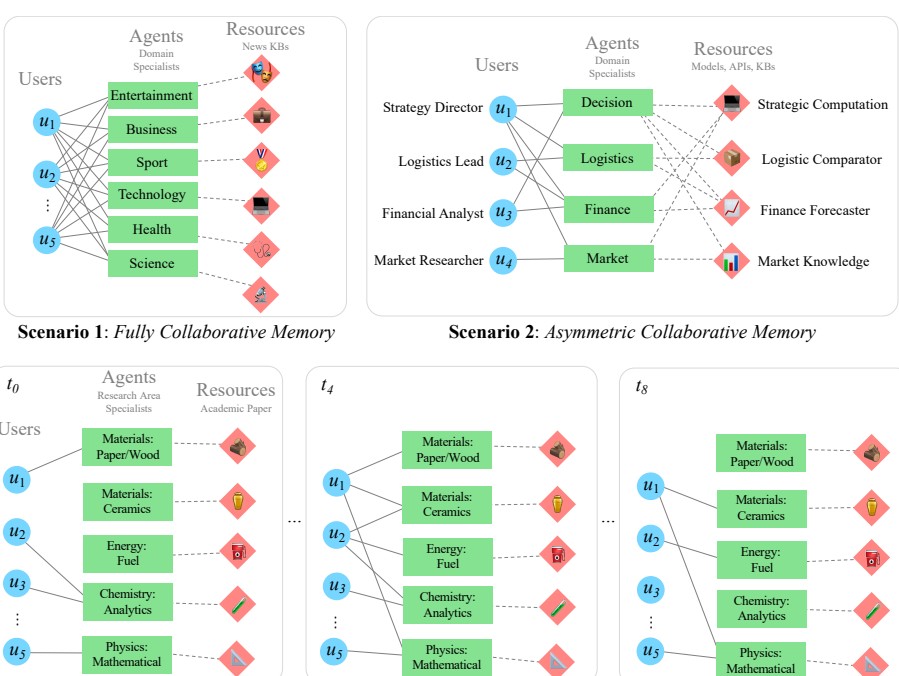

Figure 1: Illustration of multi-user, multi-agent collaboration. Scenario 1 (top-left): A fully collaborative memory environment where all users share unrestricted access to all agents. Scenario 2 (top-right): An asymmetric collaborative memory setup with heterogeneous privilege levels. Scenario 3 (bottom): Dynamically changing access, where permissions are granted or revoked over time.

However, these architectures mainly assume a single-user, single-agent paradigm with centralized, globally accessible memory. This assumption fails in many real-world applications that are inherently *multi-user* and *multi-agent*, such as collaborative enterprise assistants, multi-user productivity platforms, and distributed workflow systems Packer et al. (2023). In these settings, sharing memory across user boundaries can reduce redundant inquiries, maintain consistency, and improve collective reasoning (Figure 1). Yet research often overlooks these complexities, leaving two key challenges underexplored: **1)** *Information Asymmetry:* users have access to different agents, and agents connect to different resources—making it crucial for memory systems to enforce these asymmetries to prevent unauthorized information sharing; and **2)** *Dynamic Access Patterns:* permissions shift over time with changing roles, policies, and task needs.

These asymmetric and temporally evolving constraints pose a central question: How can a multi-user, multi-agent system maximize the utility of collective memory while ensuring that information sharing conforms to permissions governing what can be shared, when, and by whom? In this paper, we take a first step toward addressing this question by introducing a framework designed specifically for multi-user, multi-agent memory management under asymmetric and dynamic access constraints. Our proposed framework comprises several key components:

- **Dynamic bipartite access graphs.** We formalize time-dependent permissions using two bipartite graphs: one representing user-to-agent permissions, and another representing agent-to-resource permissions. These graphs explicitly encode the evolving permissions landscape, capturing real-world dynamics such as user onboarding, role changes, and evolving policy constraints.

- **Two-tier memory system.** To enable controlled yet flexible knowledge sharing, each agent maintains a dual-tier memory architecture consisting of private memory and shared memory. Private memory isolates sensitive information to individual user contexts, while shared memory enables knowledge transfer among users when allowed by the system's access policies.

- **Fine-grained read and write policies.** Memory interactions are governed by distinct read and write policies. The write policy projects raw interaction logs into structured memory fragments,

selectively allocating these fragments to either private or shared memory tiers. Conversely, the read policy dynamically constructs a memory view tailored to each agent's current permissions, selectively incorporating memory fragments according to fine-grained access constraints. Importantly, these policies are highly configurable, supporting specification and enforcement at multiple granularity levels—system-wide, agent-specific, and user-specific—and are adaptive over time.

To our knowledge, this is the first formulation of memory sharing that explicitly accounts for fine-grained access asymmetries in multi-agent, multi-user systems. Our work establishes foundational concepts and formalizations, laying the groundwork for future research into efficient, secure, and adaptive collaborative multi-agent systems.

## 2 RELATED WORK

**Long-term memory for single LLM agents.** Early work on augmenting LLMs with *persistent* context treats memory as a flat retrieval table, but recent systems introduce hierarchical or structured representations that better support long-horizon reasoning. MemGPT leverages operating-system abstractions to emulate virtual memory and extend the effective context window of an LLM Packer et al. (2023). MemTree organizes interaction history as a hierarchical dynamically evolving tree whose nodes hold abstracted summaries, enabling continual updates of memory Rezazadeh et al. (2024). HippoRAG constructs a hippocampal-inspired knowledge graph and applies Personalized PageRank to integrate new information while avoiding catastrophic forgetting Gutiérrez et al. (2024). GraphRAG takes a complementary approach, constructing an entity-relation graph over the corpora and retrieving sub-graphs to answer complex queries Edge et al. (2024). Although highly effective, all of these designs assume a *single-user, single-agent* setting with globally visible memory.

**Multi-agent collaboration and shared memory.** LLM agents increasingly collaborate as heterogeneous teams whose members specialize in search, planning, self-critique, and tool calling. AutoGen provides an open-source conversation framework that lets developers script arbitrary interaction graphs among tool-augmented agents Wu et al. (2023). AgentVerse explores emergent social behaviors when groups of agents are asked to solve tasks collectively, showing measurable gains over single-agent baselines Chen et al. (2023). MoSA (Mixture-of-Search-Agents) uses Monte-Carlo Tree Search to fuse diverse reasoning trajectories proposed by multiple LLMs and consistently outperforms strong single-model solvers on math and commonsense benchmarks Yang et al. (2025). COPPER introduces a *shared reflector* that learns to issue counterfactual rewards, mitigating credit-assignment issues in multi-agent planning Bo et al. (2024). Most of these frameworks either maintain *no* persistent memory or assume a *fully shared* memory store; Memory Sharing Gao & Zhang (2024) begins to address this gap by letting agents asynchronously contribute to, and retrieve from, a single common memory pool, but it still overlooks user-level privacy or access constraints.

**Access-control models for knowledge sharing.** The security literature offers mature abstractions for permission management. Role-Based Access Control (RBAC) assigns users to roles that carry predefined privileges, supporting efficient policy administration in large organizations Sandhu et al. (2000). Attribute-Based Access Control (ABAC) generalizes RBAC by deciding authorization through logical predicates over arbitrary subject, object, and environmental attributes Hu et al. (2013). Our framework inherits the policy modularity of ABAC while explicitly modeling *time-varying* bipartite graphs between users, agents, and resources; to our knowledge, no existing LLM memory system combines such policies with provenance-aware fragment storage.

**Summary.** Prior work establishes the benefits of (i) structured long-term memory for individual agents and (ii) division-of-labor in multi-agent systems; however, none accommodates the asymmetric, dynamic permissions that arise when *multiple human users* interact with collaborative *multiple LLM agents*. Our contribution is to bridge these threads by embedding formal access-control graphs and policy-conditioned read/write transformations directly into the memory substrate, enabling safe and auditable cross-user knowledge transfer.

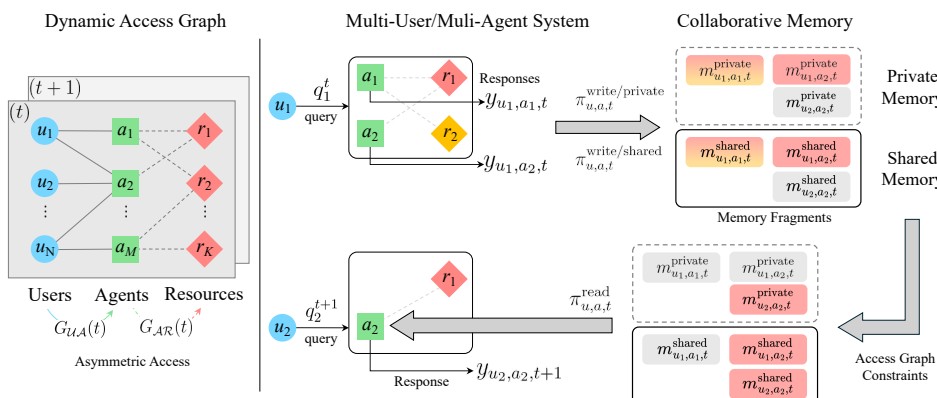

Figure 2: Left: Illustration of the dynamic multi-user environment, where multiple users collaborate with various agents that each have access to different resources. Right: Illustration of Collaborative Memory. User $u_1$ sends a query $q_1$ and receives responses from agents $a_1$ and $a_2$. These responses are passed to the write policies $\pi^{\text{write/private}}$ and $\pi^{\text{write/shared}}$, which updates both the private and shared memories. Memory fragments that are not accessible are shown in gray. When user $u_2$ issues another query $q_2$, the agent access resource $r_1$ as well as reading from the collaborative memory, retrieving only the information it is permitted to access.

## 3 COLLABORATIVE MEMORY

Large-scale, multi-user applications increasingly rely on ensembles of *LLM agents* that reason, communicate, and coordinate actions—both among themselves and with external resources such as tools, APIs, or databases Tran et al. (2025a). Prior research Zhong et al. (2023); Liu et al. (2023); Tran et al. (2025b) demonstrates that granting agents access to a persistent memory of past interactions can significantly enhance task performance; however, most existing designs assume a *single-user* setting, thus overlooking the benefits of *cross-user* knowledge transfer. In this work, we present **Collaborative Memory**, a framework that enables agents to share knowledge across multiple users without violating each user's or agent's access constraints. As illustrated in Figure 2, users' queries, agent responses, and write policies collectively maintain and manage both private and shared memories, ensuring that each agent can only read the information it is permitted to access.

### 3.1 SETUP AND DYNAMIC ACCESS GRAPHS

Let $\mathcal{U}$, $\mathcal{A}$, and $\mathcal{R}$ denote the sets of users, large language model agents, and resources (e.g., tools, APIs, a structured data source), respectively. For convenience, we also write $\mathcal{U}(\cdot)$, $\mathcal{A}(\cdot)$, and $\mathcal{R}(\cdot)$ to indicate subset operators over these sets. We use lowercase $u$, $a$, and $r$ to refer to individual users, agents, and resources.

At each timestep $t$, access permissions are captured by two bipartite graphs:

$$G_{\mathcal{UA}}(t) \subseteq \mathcal{U} \times \mathcal{A}, \, G_{\mathcal{AR}}(t) \subseteq \mathcal{A} \times \mathcal{R}, \tag{1}$$

where $(u_i, a_j) \in G_{\mathcal{UA}}(t)$ means user $u_i$ may invoke agent $a_j$ at time $t$, and $(a_j, r_k) \in G_{\mathcal{AR}}(t)$ indicates that $a_j$ can access resource $r_k$. These graphs evolve over time to reflect new users, revocations, or changes in resource permissions. We define:

$$\begin{aligned} \mathcal{A}(u, t) &:= \{ \, a \mid (u, a) \in G_{\mathcal{UA}}(t) \}, \\ \mathcal{R}(a, t) &:= \{ \, r \mid (a, r) \in G_{\mathcal{AR}}(t) \}. \end{aligned} \tag{2}$$

Hence, $\mathcal{A}(u, t)$ is the set of agents that user $u$ can invoke at time $t$, and $\mathcal{R}(a, t)$ is the set of resources agent $a$ can access at time $t$.

### 3.2 MEMORY TIERS AND PROVENANCE

We let $\mathcal{M}$ denote the set of all memory fragments stored (write) and retrieved (read) during user–agent interactions. Each fragment $m \in \mathcal{M}$ has immutable provenance attributes: its time of creation,

denoted $\mathcal{T}(m)$, the user who contributed it, $\mathcal{U}(m)$, the contributing agents, $\mathcal{A}(m)$, and the set of resources accessed during its creation, $\mathcal{R}(m)$.

In practice, users may want to keep certain memory fragments private while sharing others for potential cross-user benefits. Hence, we partition:

$$\mathcal{M} \;=\; \mathcal{M}^{\text{private}} \;\cup\; \mathcal{M}^{\text{shared}},$$

where $\mathcal{M}^{\text{private}}$ contains private fragments and $\mathcal{M}^{\text{shared}}$ contains fragments that can be shared across users. For each user $u$, we define a private store $\mathcal{M}^{\text{private}}(u,t)$, which includes all memories arising from user $u$'s past interactions with agents up until time $t$. Similarly, for each agent $a$, we can define a shared store $\mathcal{M}^{\text{shared}}(a,t)$, which includes all fragments agent $a$ generated for any user up until time $t$.

When agent $a$ serves user $u$ at time $t$, it can potentially access a broader set of memory fragments, denoted $\mathcal{M}(u,a,t)$. Formally,

$$\mathcal{M}(u,a,t) := \Big\{ m \in \mathcal{M} \;\Big|\; \mathcal{A}(m) \subseteq \mathcal{A}(u,t) \;\wedge\; \mathcal{R}(m) \subseteq \mathcal{R}(a,t) \Big\}. \tag{3}$$

This set may include:

1. Fragments from user $u$'s history with other agents in $\mathcal{A}(u,t)$;

2. Fragments created by agent $a$ itself while serving other users;

3. Fragments created by other users with agents that $u$ can invoke, subject to $a$'s resource-based permission $\mathcal{R}(a,t)$.

In this way, the system enforces agent- and resource-level constraints even while enabling cross-user and cross-agent sharing.

### 3.3 Read from the Collaborative Memory

At each timestep $t$, user $u$ submits a query $q$. Although the user can potentially access all agents in $\mathcal{A}(u,t)$, not every agent may be relevant to $q$. Let $\mathcal{A}(u,t,q)$ be the subset of agents deemed suitable for further processing of $q$.

Each agent $a \in \mathcal{A}(u,t,q)$ receives $q$ and the relevant memory (as filtered by a *read policy*), denoted $\pi_{u,a,t}^{\text{read}}$. Then, it generates the response $y_{u,a,t}$ based on the query, memory, and the resources that it can access. Formally:

$$y_{u,a,t} \;=\; a\big(q,\; \pi_{u,a,t}^{\text{read}}\big(\mathcal{M}(u,a,t)\big),\; \mathcal{R}(a,t)\big).$$

Such a read policy can, for instance, limit the number of memory fragments retrieved or filter them by specific keywords. To produce a final response to the user, a coordinator or aggregator (e.g., another LLM module) can synthesize all agent outputs $\{y_{u,a,t}\}_{a \in \mathcal{A}(u,t,q)}$ into a coherent reply.

### 3.4 Write to the Collaborative Memory

Upon producing an output $y_{u,a,t}$, the system applies a *write policy* to insert new information into the memory. We define two kinds of write policies:

$$\pi_{u,a,t}^{\text{write/private}} \;:\; y_{u,a,t},\; \mathcal{M}^{\text{private}}(u,t) \;\mapsto\; \mathcal{M}^{\text{private}}(u,+1) \tag{4}$$

$$\pi_{u,a,t}^{\text{write/shared}} \;:\; y_{u,a,t},\; \mathcal{M}^{\text{shared}}(u,t) \;\mapsto\; \mathcal{M}^{\text{shared}}(u,+1) \tag{5}$$

By separating private and shared write policies, users can customize the level of confidentiality applied to their contributions. For example, these policies may anonymize entities, redact sensitive information, or block certain kinds of content from being stored. The write policies can also range from straightforward to highly sophisticated. A direct implementation could simply embed the output $y_{u,a,t}$ and insert the resulting vector into a storage table. More advanced approaches may incorporate richer memory management structures Rezazadeh et al. (2024) to handle structured knowledge abstractions.

**Granularity and Dynamics** Both the read policy $\pi_{u,a,t}^{\text{read}}$ and the write policies $\pi_{u,a,t}^{\text{write}/*}$ can be tuned at various scopes: $\pi_{u,a,t}^* \in \{\pi_{\text{global}}^* \text{ (shared by all)}, \pi_u^* \text{ (per user)}, \pi_a^* \text{ (per agent)}, \pi_t^* \text{ (over time)}\}$. This flexible design lets system administrators and users independently fine-tune which information is surfaced to each agent (*read time*) and which is persisted back into the memory (*write time*), enabling rich forms of collaboration while maintaining strict access control.

## 4 IMPLEMENTATION DETAILS

**Models** All LLM-based components—the task coordinator, domain-specialized agents, the *read*/*write* memory-transformation modules, and the response aggregator—use `gpt-4o`. Vector embeddings are generated with `text-embedding-3-large`. Agents interact with external tools through the OpenAI function-calling interface.

**Multi-Agent Interaction Loop** The coordinator LLM receives (i) the user query, (ii) the natural-language specialization string for every agent accessible by the user, and (iii) the conversation history. At each round it outputs a JSON object of the form `{"agent": ID, "subquery": ...}` and may terminate early by outputting `{"stop": true}`.

**Resources** External resources are exposed as Python callables with JSON schemas—for example, a knowledge-base search defined as `query → top_k_results`. The underlying LLM performs function-calling to infer the arguments and invokes these functions via the standard OpenAI interface.

**Memory Encoder** After completing a subtask, the conversational trace are mapped into candidate LLM-generated key–value fragments. Each fragment is annotated with provenance denoting contributing agents, resources used, and creation time.

**Memory Retrieval** For an incoming subquery, we retrieve (a) the top-$k_{\text{user}}$ fragments from the user-specific tier and (b) the top-$k_{\text{cross}}$ fragments from the cross-user tier that satisfy the provenance constraint. Cosine similarity between the subquery embedding and fragment keys guides retrieval.

**Policy Instantiation** We consider two instantiations of the read/write policies: (1) *Simple*—the read policy returns admissible fragments verbatim and the write policy stores generated fragments without modification; and (2) *Transformation*—global, and optionally agent- or user-level, system prompts instruct an LLM to redact, anonymize, or paraphrase fragments before they are surfaced (read) or persisted (write). We adopt the simple read policy and the transformation write policy across all experiments.

**Response Aggregation** Once the coordinator halts, an aggregator LLM receives the original query together with the ordered list of (subquery, response) pairs produced by agents and synthesizes the final answer.

**Metrics** We track three performance metrics aggregated over evaluation phases. **Accuracy** measures the average normalized correctness of system responses against ground truth. **Agent Utilization** captures the mean number of distinct agents invoked per query, reflecting coordination overhead. Finally we track **Resource Utilization** is defined as the mean number of knowledge-base or API calls per query, serving as an indirect yet measurable proxy for overall system efficiency.[1]

## 5 EXPERIMENTAL EVALUATION

We evaluate our framework under three progressively complex scenarios, each highlighting a distinct dimension of collaborative memory management. In all cases, we measure accuracy, resource consumption, and privacy compliance to demonstrate the versatility and robustness of our approach.

- **Scenario 1 (Fully Collaborative Memory):** All users have unrestricted access to a global memory pool, illustrating how cross-user collaboration can reduce computational overhead.

- **Scenario 2 (Asymmetric Collaborative Memory):** Users have heterogeneous privilege levels restricting their visibility into resources and memory fragments. We show that even partial collaboration yields efficiency gains while safeguarding private data.

---

[1]Agents such as deep research from OpenAI can exhibit significant latency fluctuations (5–30 minutes) OpenAI (2025), making direct latency assessments challenging.

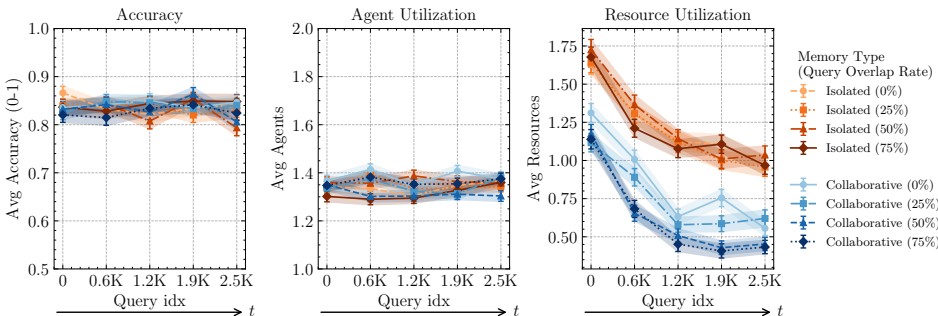

Figure 3: Scenario 1 (Fully Collaborative Memory). Performance of `MultiHop-Rag` under varying degrees of query overlap among five users. Both isolated and collaborative memory exhibit similar accuracy and agent utilization. However, as the number of queries increases (while remains confined to the same six domains), resource usage declines for both approaches—yet collaborative memory consistently achieves a more substantial reduction across all degrees of query overlap.

- **Scenario 3 (Dynamically Evolving Collaborative Memory):** Permissions are granted or revoked in real time, testing the system's ability to adapt while maintaining strict confidentiality guarantees.

## 5.1 SCENARIO 1: FULLY COLLABORATIVE MEMORY

*Cross-user collaborative memory significantly reduces overhead and improves efficiency.*

**Task** We use the MultiHop-RAG dataset Tang & Yang (2024), which contains 609 English news articles spanning six domains (technology, entertainment, sports, science, business, health). The dataset includes 2,556 multi-hop questions requiring inference, comparison, and temporal reasoning. We measure accuracy by comparing answers to ground truth with an LLM-based judge.

To simulate multiple resources and domain experts, we partition the corpus into six domain-aligned knowledge bases: `entertainment_kb`, `business_kb`, `sports_kb`, `technology_kb`, `health_kb`, and `science_kb`. We then deploy six domain-specialist agents (*entertainment_agent*, *business_agent*, *sports_agent*, *technology_agent*, *health_agent*, *science_agent*), each granted exclusive access to one resource. Five users have full permissions to query any agent.

To introduce realistic multi-user query overlaps, we apply KMeans clustering to group queries into 10 clusters. From each cluster, a subset is sampled to form a set of "global queries" shared by all users. The remaining queries in each cluster are uniquely assigned to individual users.

**Results** Figure 3 shows average accuracy, agent utilization, and resource utilization over time. Under fully collaborative memory, accuracy remains above 0.90 on average across all query indices. In contrast, isolated memory configurations exhibit slightly lower accuracy, especially as the overlap rate increases. Collaboration also significantly improves resource utilization. Resource usage decreases by up to 61% at 50% overlap and by 59% at 75% overlap compared to isolated memory. To assess robustness across model capacities for both generation and retrieval, we ablated larger vs. smaller LLMs and embedding models (Appendix § C.5). Collaborative Memory saved resources consistently with similar accuracy across capacities, showing gains are not model-size specific.

## 5.2 SCENARIO 2: ASYMMETRIC COLLABORATIVE MEMORY

*Partial collaboration increases efficiency while respecting different privilege levels.*

**Task** We curate a synthetic dataset of 200 business project queries, each mapped to a distinct high-level objective (e.g., market analysis, logistics planning, or financial forecasting). Example queries range from "Identify high-demand skincare markets in Europe" to "Craft a strategic growth plan for smart home devices in privacy-regulated markets." To address these queries, we simulate four user roles—Market Researcher, Financial Analyst, Logistics and Operations Lead, and Strategy

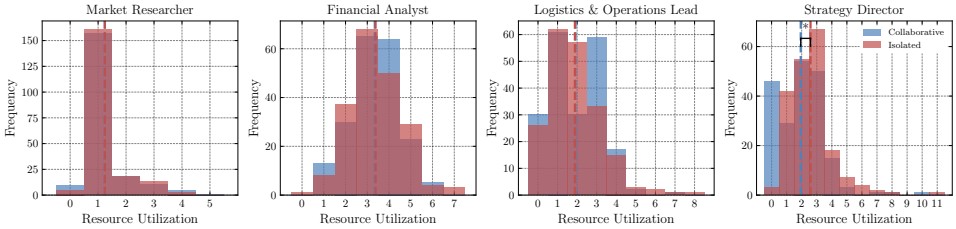

Figure 4: Scenario 2 (Asymmetric Collaborative Memory). Resource usage with and without asymmetric collaboration. Even limited cross-user visibility leads to fewer agent or knowledge-base calls than a completely isolated configuration.

Director—each with unique expertise. In parallel, we deploy four specialized LLM agents, each granted access to different resources (market databases, financial forecasts, logistics comparators, and strategic computation). Access is asymmetric: only the Strategy Director can consult all agents for a comprehensive view, while other roles are limited to subsets of the agents.

**Results**   Since no ground truth answers are available for these open-ended questions, we focuses on resource utilization in the evaluation. Figure 4 shows a histogram of resource usage per user. Compared to an isolated setup with no memory sharing, asymmetric collaboration reduces overall resource calls. Intermediate insights discovered by one user can flow to another user with matching privileges, reducing repeated queries to the same tools or knowledge bases. By the time the Strategy Director (with the highest privileges) consolidates final recommendations, partial sharing has already eliminated a significant portion of redundant work. We also quantify the cost of policy enforcement at different granularities (Appendix § D.4). Adding system-, agent-, and user-level controls introduces only negligible overhead—total LLM-call counts stay within statistical variation across roles—supporting practical fine-grained access control.

## 5.3   SCENARIO 3: DYNAMICALLY EVOLVING COLLABORATIVE MEMORY

*Adaptive permissions allow real-time updates without compromising security.*

**Task**   We use the SciQAG dataset Wan et al. (2024), a scientific question-answering benchmark spanning fields such as biology, chemistry, and physics. Five scientific categories are selected, each serving as a distinct resource. We deploy five RAG agents, each providing access to one category's corpus. There are five users, and each user submits 4 queries per category, for a total of 100 queries.

To simulate dynamic access control, we represent users and resources as nodes in an evolving graph, where edges denote granted access. Starting with an empty graph, we iteratively add edges (i.e., grant permissions) via Bernoulli trials until the graph reaches 5 edges. We label these discrete time steps $t_0, t_1, \ldots$ at intervals of 100 queries. Specifically:

- At $t_0$, we begin with a sparse graph.
- From $t_1$ to $t_4$, we progressively add more edges (up to 25 total), representing increased access.
- From $t_5$ to $t_8$, edges are revoked step by step, mirroring real-world privilege withdrawal.

The same 100 queries are used in each phase, but agents depend on the active access graph.

**Results**   Figure 5 summarizes the results under dynamically changing permissions. Accuracy rises as additional access is granted and declines as privileges are revoked, demonstrating the strong coupling between resource availability and response quality. Agent usage similarly increases when users gain access to more agents and decreases upon revocation.

Notably, the average number of queried resources tends to drop over time. This behavior reflects the memory mechanism's ability to reuse previously retrieved information (i.e., stored memory fragments) instead of repeatedly querying external resources. The access matrix in Figure 6 further confirms that users only access agents and resources explicitly granted by the graph, ensuring strict adherence to access control policies. For scalability, we expanded from 5 to 50 users with proportionally scaled permissions (Appendix § E.5). Execution time rose by only  10% despite a 10× increase in

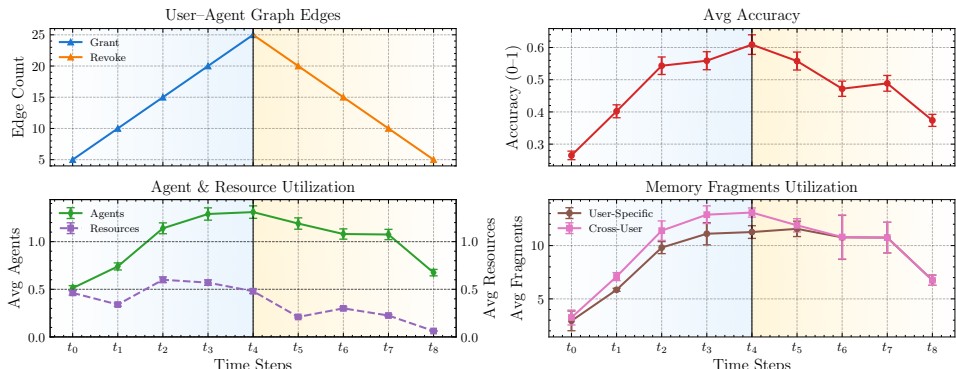

Figure 5: Scenario 3 (Dynamically Evolving Collaborative Memory). System performance over eight time blocks with dynamically changing privileges. Accuracy (top) tracks the available resources, while agent (middle) and resource (bottom) usage also shift in response to access graph updates.

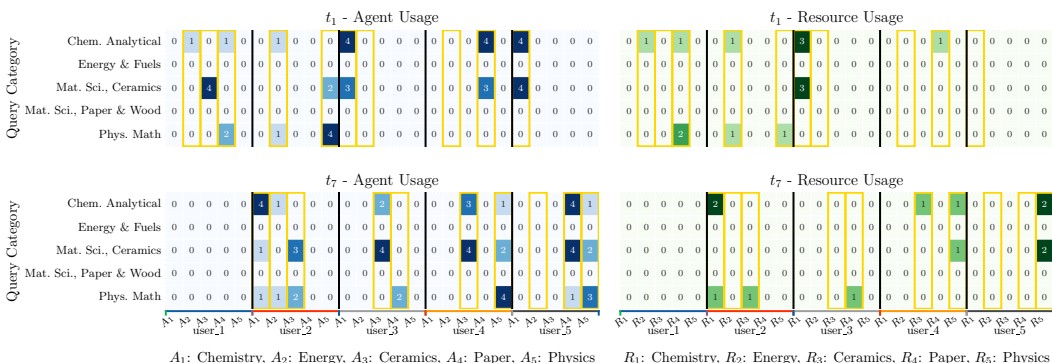

Figure 6: Scenario 3 (Dynamically Evolving Collaborative Memory). Agent and resource usage across user queries from different categories. Yellow rectangles indicate granted access, with values representing the corresponding usage counts.

complexity. Compared with Memory Sharing Gao & Zhang (2024) baseline (Appendix § E.6), our framework matches task quality while eliminating leakage entirely (0% vs. 59.03%), underscoring the value of explicit access control.

# 6 LIMITATIONS

Collaborative Memory enables multi-user collaboration, but several limits remain: (1) Privacy and regulatory barriers hinder large-scale real-world data, so we use benchmarks or synthetic queries that may miss full complexity; in this work, we take the first step to formalize the problem setting and establish a protocol, but richer real-world benchmarks are still needed. (2) Our experiments focus on controlled settings with moderate user/agent counts, leaving high-concurrency, fast-evolving enterprise scenarios underexplored. (3) Production resource evaluation is difficult under unpredictable API latencies; here we proxy it with resource call counts.

# 7 CONCLUSIONS

In this paper, we drew on Distributed Cognition to tackle collaborative memory management in real-world multi-user, multi-agent scenarios. Our framework, Collaborative Memory, addresses information asymmetry and dynamic access constraints by integrating dynamic bipartite access graphs, a two-tier memory system, and fine-grained read/write policies. This design supports secure, context-aware knowledge sharing that respects evolving permissions. In three progressively complex scenarios, we observe significantly lower resource utilization than in single-user baselines, while maintaining strict privacy compliance.

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

APPENDIX ARRANGEMENT

The Appendix is organized as follows.

- **Section § A: Discussion of the Broad Impact of Collaborative Memory**
- **Section § B: From Query to Response**
  - **Section § B.1**: The detailed pipeline.
  - **Section § B.2**: The read/write policies.
- **Section § C: Scenario 1: Fully Collaborative Memory**
  - **Section § C.1**: Dataset details, including description and statistics.
  - **Section § C.2**: Experimental setup, including:
    * Access Graph Configuration: Description of the access graph used.
    * Agent/Knowledge Base Configuration: Description of agents, resources, coordinators, and aggregators.
  - **Section § C.3**: Examples of read/write interactions.
  - **Section § C.4**: Performance breakdown by query type.
  - **Section § C.5**: Model scale ablation study.
- **Section § D: Scenario 2: Asymmetric Collaborative Memory**
  - **Section § D.1**: Dataset details, including dataset generation.
  - **Section § D.2**: Experimental setup, including:
    * Access Graph Configuration: Description of the access graph used.
    * Agent and Knowledge Base Configuration with Examples: Includes descriptions of agents, resources, coordinators, and aggregators, along with illustrative examples of agent and resource input/output.
  - **Section § D.3**: Examples of read/write interactions.
  - **Section § D.4**: Policy granularity overhead analysis.
- **Section § E: Scenario 3: Dynamically Evolving Collaborative Memory**
  - **Section § E.1**: Dataset details, including description and statistics.
  - **Section § E.2**: Experimental setup, including:
    * Access Graph Configuration: Description of the dynamic access graph used and how they are generated.
    * Agent/Knowledge Base Configuration: Description of agents, resources, coordinators, and aggregators.
  - **Section § E.3**: Examples of read/write interactions.
  - **Section § E.4**: Raw performance data and complete access matrix under different access graph configurations.
  - **Section § E.5**: System scalability analysis.
  - **Section § E.6**: Comparison with Memory Sharing baseline.

## A    DISCUSSION OF THE BROAD IMPACT OF COLLABORATIVE MEMORY

This paper proposes *Collaborative Memory*, a modular framework for permission-aware memory sharing in multi-agent, multi-user systems. Our key contributions include:

- A novel formulation of access-asymmetric memory sharing, explicitly modeling time-varying user-agent-resource permissions using dynamic bipartite graphs.

- A two-tier memory architecture that separates private and shared memory to balance knowledge isolation with controlled collaboration.

- Configurable read and write policies that support fine-grained, adaptive access control at system, user, or agent levels.

- A decoupled and extensible design, allowing seamless integration with alternative memory systems (e.g., MemTree).

As multi-agent systems are increasingly deployed in real-world applications, the need for effective coordination and secure information sharing has become paramount. Collaborative Memory addresses these challenges by enabling shared memory to improve collaboration and efficiency while enforcing strict controls over access. By enhancing both collaborative capability and information safety, this framework lays the foundation for building scalable, trustworthy multi-agent AI systems across diverse domains. We demonstrate its applicability through three representative scenarios: (1) Fully Collaborative Memory, (2) Asymmetric Collaborative Memory, and (3) Dynamically Evolving Collaborative Memory.

## B    QUERY TO RESPONSE DETAILS

### B.1    PIPELINE

Figure 7 illustrates the detailed pipeline from a user query to the final response in our multi-agent memory-sharing framework, as outlined below:

- **Query Submission**: When user $u_1$ submits a query $q_1$, the pipeline is triggered within the multi-agent memory-sharing framework.

- **Agent Selection | Coordinator**: The coordinator selects the next agent based on the query content and the access graph, ensuring that only authorized agents are eligible to respond.

- **Response Generation | Agents**:

  - The selected agent gathers relevant memory fragments and accessible resources.

  - It generates a response, denoted as $y_{u_1,a_1,t}$ (in the case of agent $a_1$).

  - The response is used to update the collaborative memory.

- **Aggregation | Aggregator**: After all eligible agents have responded, their intermediate outputs are aggregated by the aggregator to produce the final response.

The specific configurations of the coordinators and aggregators will be described in the corresponding scenario settings. Both the intermediate and final responses can be used to update the collaborative memory. In our current implementation, we update the memory using the intermediate responses. We will publicly release our code and dataset to support transparency and future research.

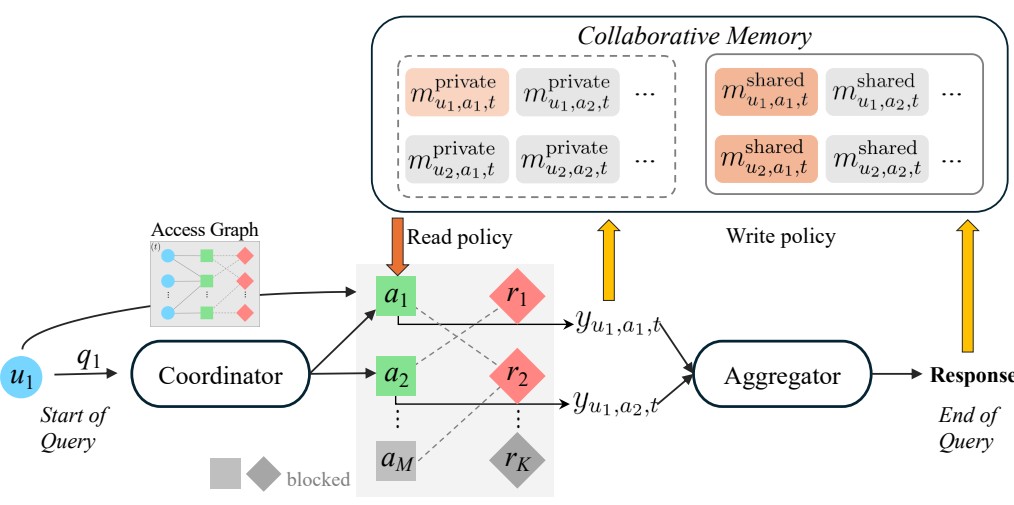

Figure 7: From user query to response: the detailed pipeline. The *Coordinator*, and *Aggregator* are implemented using the GPT-4o API.

## B.2 READ/WRITE POLICY

For the read policy, we adopt a straightforward strategy that combines both private and shared memory fragments accessible to the user. For the write policy, we employ a system-level transformation applied uniformly across all users by leveraging LLMs with dedicated prompts. When writing to private and shared memory, we transform the memory with the different prompts as presented in Table 1. While we begin our exploration using this unified policy, the framework is inherently flexible and can be extended to support more fine-grained policies—customized at the level of individual users or agents as needed.

Table 1: System-Level Prompts for Memory Writing Policies

| Memory Type | Prompt |
|---|---|
| Private Memory | *Extract key concepts from interactions that would be useful for the specific user. Focus on creating standalone memories that capture core information. Format each memory as a clear key-value pair where the key is a concise query or topic and the value is a comprehensive answer or explanation.* |
| Shared Memory | *Extract generally applicable knowledge from interactions that would benefit any user. Focus on creating shareable memories that contain universal information. Remove any user-specific details or personalized examples. Format each memory as a clear key-value pair where the key is a concise query or topic and the value is a comprehensive answer or explanation.* |

## C SCENARIO 1: FULLY COLLABORATIVE MEMORY

### C.1 DATASET DETAILS

The following table provides a comprehensive overview of the MULTIHOP-RAG dataset used in our experiments. We employ 609 news articles with an average length of approximately 2,046 tokens. From these, we derive 2,556 multi-hop queries spanning inference, comparison, temporal reasoning, and null-query instances. Table 2 reports the detailed breakdown of document counts, average lengths, and query-type distributions.

Table 2: Statistics of the MULTIHOP-RAG dataset.

| Statistic | Count | Percentage |
|---|---|---|
| **Documents** | | |
| Total articles | 609 | — |
| Average tokens per article | 2,046 | — |
| **Queries (N = 2,556)** | | |
| Inference queries | 816 | 31.9% |
| Comparison queries | 856 | 33.5% |
| Temporal queries | 583 | 22.8% |
| Null queries | 301 | 11.8% |

### C.2 EXPERIMENTAL DETAILS

We evaluate four multi-user datasets with global query overlap rates $\rho \in \{0\%, 25\%, 50\%, 75\%\}$. For each dataset, we run two configurations: *Shared Memory* (cross-user memory sharing enabled) and *Isolated Memory* (sharing disabled). Memory retrieval is configured with a user-specific tier size $k_{\text{user}} = 10$, a shared tier size $k_{\text{cross}} = 10$.

#### C.2.1 ACCESS GRAPH CONFIGURATION

All five users (user_1 through user_5) have permission to invoke each of the six domain-specialist agents. Conversely, each agent is restricted to a single, domain-aligned resource. These fixed access graphs—complete user-to-agent and one-to-one agent-to-resource—remain constant throughout the experiments to isolate the effects of memory sharing.

#### C.2.2 AGENT/KNOWLEDGE BASE CONFIGURATION

We deploy six specialist agents, each paired with a dedicated knowledge base. Table 3 lists each agent's description and full system prompt. The corresponding resources contain the following number of documents: entertainment (114), business (81), sports (211), technology (172), health (10), and science (21), each indexed with precomputed embeddings.

Table 3: Domain agents and their system prompts in the fully collaborative scenario.

| Agent | Description | System Prompt |
|---|---|---|
| entertainment_agent | Specialist in entertainment news—films, TV, music, celebrity culture, and pop-culture trends. | You are an Entertainment News specialist with deep expertise in:
• Summarizing the latest developments in film, television, and music
• Contextualizing celebrity news and red-carpet events
• Tracking pop-culture trends and fandom reactions
• Providing concise, engaging overviews of entertainment topics
Always check relevant memories first. When those are insufficient, use the available tools. Prioritize information from memories and tools. |
| business_agent | Specialist in business news—markets, companies, economics, and finance. | You are a Business News specialist with deep expertise in:
• Reporting market movements and stock-market summaries
• Explaining corporate earnings, M&A, and leadership changes
• Analyzing economic indicators and policy developments
• Providing clear explanations of financial concepts and trends
Always check relevant memories first. When those are insufficient, use the available tools. Prioritize information from memories and tools. |
| sports_agent | Specialist in sports news—games, scores, athletes, and tournament analysis. | You are a Sports News specialist with deep expertise in:
• Summarizing match results and key highlights
• Interpreting player statistics and performance metrics
• Previewing upcoming fixtures and tournament storylines
• Offering tactical analysis and historical context
Always check relevant memories first. When those are insufficient, use the available tools. Prioritize information from memories and tools. |
| technology_agent | Specialist in technology news—gadgets, innovations, startups, and industry trends. | You are a Technology News specialist with deep expertise in:
• Covering product launches and tech reviews
• Analyzing startup ecosystems and funding rounds
• Tracking breakthroughs in AI, software, and hardware
• Explaining complex technical concepts in accessible terms
Always check relevant memories first. When those are insufficient, use the available tools. Prioritize information from memories and tools. |

*Continued on next page*

Table 3: Domain agents and their system prompts in the fully collaborative scenario. (continued)

| Agent | Description | System Prompt |
|-------|-------------|---------------|
| health_agent | Specialist in health news—medical research, public health, wellness, and policy. | You are a Health News specialist with deep expertise in:
• Summarizing clinical studies and medical breakthroughs
• Explaining public-health announcements and guidelines
• Contextualizing wellness and fitness trends
• Clarifying healthcare policy and its societal impact
Always check relevant memories first. When those are insufficient, use the available tools. Prioritize information from memories and tools. |
| science_agent | Specialist in science news—discoveries in physics, biology, environment, and research. | You are a Science News specialist with deep expertise in:
• Reporting on breakthroughs across physical and life sciences
• Interpreting new research findings and methodologies
• Explaining fundamental scientific concepts clearly
• Exploring implications for technology, environment, and society
Always check relevant memories first. When those are insufficient, use the available tools. Prioritize information from memories and tools. |

C.3 EXAMPLES

To illustrate the direct impact of persistent memory on resource efficiency, Table 4 presents two "write–then–read" examples. Each example details a memory fragment's unique identifier, key–value content, the query that generated it, and the subsequent query that retrieved it. These cases show that later questions can be answered by recalling existing fragments—without issuing new knowledge-base searches—thereby eliminating redundant external calls and reducing overall resource usage.

Table 4: Memory write–read examples for user_3 in the fully collaborative scenario, showing both user-specific and cross-user entries.

**Example 1 (Private Memory)**
*Discussion: This private fragment captures TechCrunch's critical perspective on Meta by summarizing two related reports, enabling the same user to answer a subsequent AI-related query with richer context.*

| | |
|---|---|
| **Memory ID** | 9bea41d8-7d5a-430a-bd5d-a0bf42a3128f |
| **Key** | **TechCrunch**'s perspective on Meta |
| **Value** | **TechCrunch**'s reports on Meta from October 19, 2023, and November 30, 2023, demonstrate a consistent critical perspective towards Meta's policies and practices. The reports focus on moderation issues affecting Palestinian voices and the legality and fairness of Meta's ad-free subscription service, highlighting concerns about free expression, consumer rights, and regulatory compliance. |

| | |
|---|---|
| **Generated at** | Query #51 (user_3): *After **TechCrunch** reported on Meta's moderation issues affecting Palestinian voices on October 19, 2023, and again on Meta's ad-free subscription service being potentially illegal and unfair on November 30, 2023, was there consistency in the news source's critical perspective towards Meta's policies and practices?* |
| **Used at** | Query #54 (user_3): *Who is the Silicon Valley figure associated with the rise of artificial intelligence, mentioned in articles by 'The Age', 'Fortune', and 'TechCrunch', who faced no removal efforts by Anthropic co-founders, has been described as generous and impactful, yet is also theorized to have had transparency issues with the board?* |

**Example 2 (Shared Memory)**

*Discussion: Highlights cross-user sharing, with a fragment about Sam Bankman-Fried's criminal trial about cryptocurrency generated by user_2 and reused by user_3 for a related financial question.*

| | |
|---|---|
| **Memory ID** | 01d1b5d4-0f17-42c6-903c-e40e572019a4 |
| **Key** | Who is facing a criminal trial related to **cryptocurrency** fraud? |
| **Value** | Sam Bankman-Fried, the former CEO of the **crypto** exchange FTX, is facing a criminal trial. He was once considered the trustworthy face of the **cryptocurrency** industry and is accused of committing fraud for personal gain. |
| **Generated by** | Query #10 (user_2): *Who is the individual facing a criminal trial, as reported by both TechCrunch and Fortune, who was once considered the trustworthy face of the **cryptocurrency** industry according to The Verge, and is accused by the prosecution of committing fraud for personal gain?* |
| **Used by** | Query #15 (user_3): *Who is the individual associated with the **crypto** exchange FTX, who has been accused of using customer funds for a buyout and is facing multiple charges of fraud and conspiracy, as reported by sources like Fortune, The Verge, and TechCrunch?* |

## C.4 PERFORMANCE BREAKDOWN BY QUERY TYPE

Figure 8 breaks down performance by query type—(top row) inference, (middle) comparison, and (bottom) temporal—across three key metrics: average accuracy (left), agent utilization (center), and resource utilization (right). In each plot, solid lines show the shared-memory condition and dashed lines the isolated-memory condition; colors distinguish global query overlap rates (0%, 25%, 50%, and 75%). Horizontal red lines in the accuracy panels indicate the human-annotated evidence upper-bounds for each task.

For inference queries, both configurations achieve near-ceiling accuracy, but shared memory steadily reduces the number of agent calls and external retrievals, particularly at higher overlap rates. Comparison tasks exhibit a small accuracy gain under sharing, with a more pronounced drop in resource usage as overlap increases. Temporal queries, which are hardest, benefit most in resource efficiency from shared memory, while maintaining comparable accuracy and modest reductions in agent invocations.

## C.5 MODEL SCALE ABLATION STUDY

We investigate the robustness of collaborative memory across different model architectures to establish generalizability beyond large-scale language models. This analysis compares performance between GPT-4o with text-embedding-3-large versus GPT-4o-mini with text-embedding-3-small configurations.

Table 5 presents resource utilization metrics across model configurations and query overlap rates. While smaller models exhibit higher absolute resource consumption—consistent with reduced per-query capabilities—the relative efficiency gains from collaborative memory remain substantial. These results confirm that the framework's core value proposition—leveraging cross-user memory sharing for computational efficiency—is preserved across diverse model architectures.

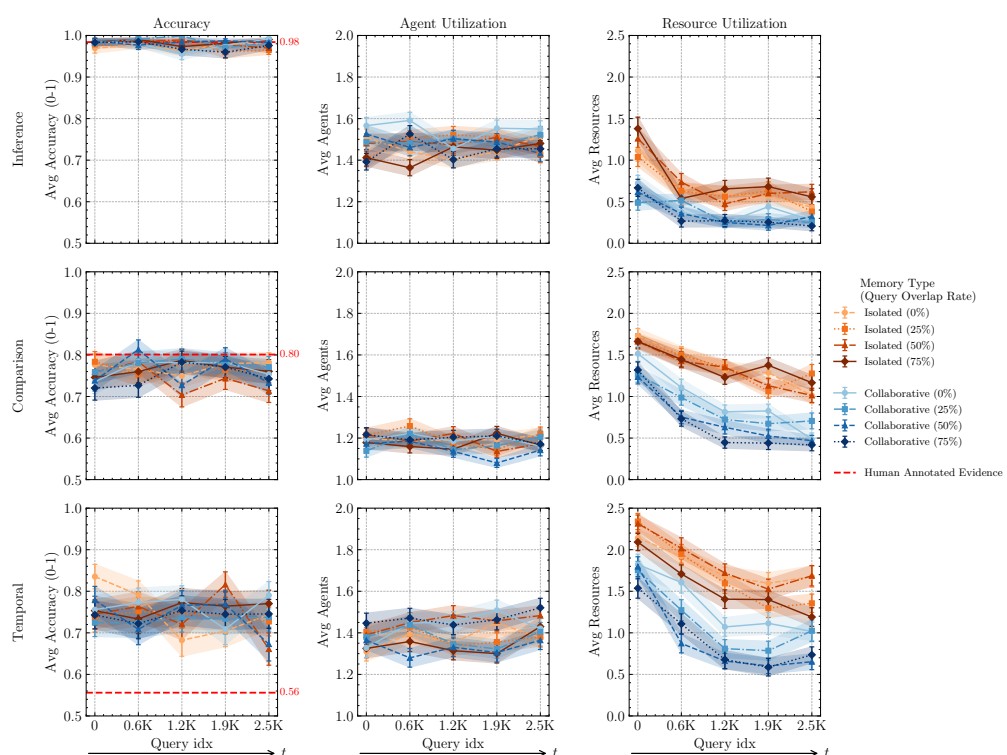

Figure 8: Performance by query type over five time bins: (left) average accuracy, (center) average agents per query, and (right) average external retrievals per query.

Table 5: Resource utilization under model scale ablation across varying query overlap rates.

| | **GPT-4o** | | | **GPT-4o-mini** | | |
|---|---|---|---|---|---|---|
| **Overlap** | **Collab** | **Isolated** | $\Delta$ **(Collab – Iso)** | **Collab** | **Isolated** | $\Delta$ **(Collab – Iso)** |
| 0% | $0.83 \pm 0.02$ | $1.31 \pm 0.03$ | $-0.48^*$ | $1.24 \pm 0.03$ | $1.64 \pm 0.03$ | $-0.40^*$ |
| 25% | $0.73 \pm 0.02$ | $1.31 \pm 0.03$ | $-0.57^*$ | $1.10 \pm 0.03$ | $1.68 \pm 0.03$ | $-0.58^*$ |
| 50% | $0.65 \pm 0.02$ | $1.25 \pm 0.03$ | $-0.60^*$ | $0.95 \pm 0.03$ | $1.63 \pm 0.03$ | $-0.69^*$ |
| 75% | $0.57 \pm 0.02$ | $1.32 \pm 0.03$ | $-0.75^*$ | $0.85 \pm 0.03$ | $1.60 \pm 0.03$ | $-0.75^*$ |

$^* p < 0.05$. Values represent average resource calls per query.

# D SCENARIO 2: ASYMMETRIC COLLABORATIVE MEMORY

This scenario simulates a collaborative reasoning environment involving multiple users and agents operating under access constraints and shared memory policies. The system comprises four users, each assigned a predefined role: *market researcher*, *financial analyst*, *logistics and operations lead*, and *strategy director* (user 4). These roles determine the scope of interaction with specialized agents—for example, the market researcher is restricted to delegating subtasks to the `market_agent`, while the strategy director, who has the highest level of access, can interact with all four agents across domains. This role-based routing ensures efficient task decomposition, policy enforcement, and targeted agent utilization. Figure 9 illustrates the system pipeline for Scenario 2, showcasing the coordinated collaboration among users, agents, and shared resources in response to a complex, cross-domain query. Further implementation details are provided in the following sections.

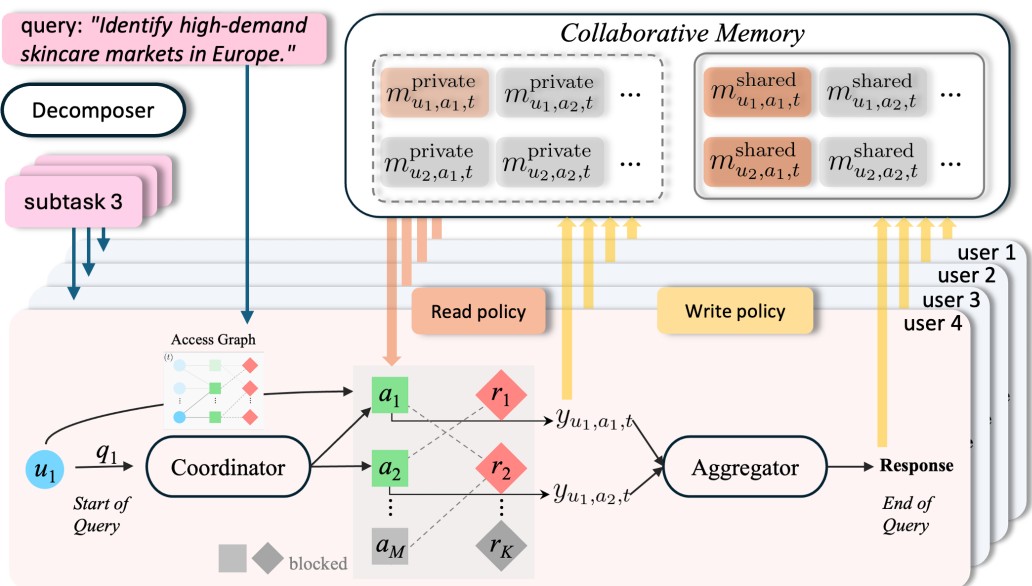

Figure 9: System pipeline for Scenario 2, illustrating the collaboration among users, agents, and resources in response to a complex query. The *Decomposer*, *Coordinator*, and *Aggregator* are implemented using the GPT-4o API.

## D.1 DATASET DETAILS

We use GPT-4o to synthesize a dataset designed to simulate multi-user, multi-agent collaborative experiments. The dataset consists of 200 business project scenarios, evenly split between 100 easy and 100 hard queries. The prompt used to generate the dataset and an example of generated dataset are shown below.

```
Prompt = f"""
Generate a JSON object describing a realistic {difficulty.upper()} business project scenario.

The format should be:

"{project_id}": {{
  "query": "Realistic business objective (e.g., enter Latin American e-commerce, assess demand
      for AI services in Asia)",
  "users": [
    {"user_id": "user_1", "role": "market"},
    {"user_id": "user_2", "role": "finance"},
    {"user_id": "user_3", "role": "logistics"},
    {"user_id": "user_4", "role": "decision_maker"}
  ]
}}
```

```
Output ONLY valid JSON. No explanation. Avoid markdown syntax.
"""
```

```
Dataset example:
"project_easy_01": {
    "query": "Identify high-demand skincare markets in Europe.",
    "users": [
        {
          "user_id": "user_1",
          "role": "market"
        },
        {
          "user_id": "user_2",
          "role": "finance"
        },
        {
          "user_id": "user_3",
          "role": "logistics"
        },
        {
          "user_id": "user_4",
          "role": "decision_maker"
        }
    ]
  }
```

## D.2 EXPERIMENTAL DETAILS

We design two types of experiments to evaluate our system: one with memory sharing and one without, memory retrieval is configured with a user-specific tier size $k_{\text{user}} = 20$, a shared tier size $k_{\text{cross}} = 20$. To simulate realistic multi-user, multi-agent interactions, we initialize four distinct users, each with specific role descriptions. These users are assigned varying levels of access to a set of specialized agents. Each agent, in turn, has controlled access to a set of task-relevant resources, forming a hierarchical access structure that mirrors real-world organizational settings.

The agent pool consists of four domain-specific agents: `market_agent`, `finance_agent`, `logistics_agent`, and `decision_agent`. Each agent is guided by a domain-specific system prompt that reflects its area of expertise. These agents interact with task-relevant resources simulated by the GPT-4o API, including a market knowledge base (`market_kb`), a financial forecast model (`finance_forecaster`), a logistics planning tool (`logistics_comparator`), and a strategic reasoning module (`strategic_computation`). For each agent and resource, we provide a representative query-response example to illustrate their behavior and capabilities. Detailed descriptions of the agents and resources can be found in Table 8 and Table 10.

```
market_agent:
Query:
Conduct a comprehensive analysis of consumer skincare trends and preferences in various
    European countries to identify potential high-demand markets.

Response:
Based on the analysis of consumer skincare trends and preferences in various European
    countries, we can identify potential high-demand markets. Here are the insights:

1. **Germany**: Consumers are increasingly interested in health and wellness, sustainability,
    and eco-friendly products. The market shows a strong preference for personalized skincare
    solutions and tech-driven products.
2. **Sweden**: ...
3. **France**: ...
4. **Italy**: ...
5. **UK**: ...
6. **Netherlands**: ...
7. **Switzerland**: ...

These countries reflect high-demand markets due to their specific trends and consumer
    preferences. They align with broader European trends such as sustainability, minimalism,
    personalization, and clean beauty, indicating potential profitability for skincare
    products that cater to these preferences.
```

```
finance_agent:
Query:
```

```
Analyze the economic indicators and purchasing power in Germany and other identified high-
    demand European markets to assess the financial viability and potential profitability of
    entering these skincare markets.

Response:
To assess the financial viability and potential profitability of entering high-demand skincare
    markets in Europe, we need to analyze the economic indicators and purchasing power in
    Germany and other identified markets such as Sweden, France, Italy, the UK, the
    Netherlands, and Switzerland.

### Economic Indicators and Purchasing Power Analysis:

1. **Germany**:
   - **GDP Growth**: Stable growth rate around 1-2%.
   - **Inflation Rate**: Stabilizing around 2-3%.
   - **Unemployment Rate**: Low, enhancing consumer spending capacity.
   - **Purchasing Power**: High disposable income, indicating potential for premium product
       sales.
2. **Sweden**: ...
3. **France**: ...
4. **Italy**: ...
5. **UK**: ...
6. **Netherlands**: ...
7. **Switzerland**: ..

### Financial Viability and Profitability Assessment:

- **High Disposable Income**: Countries like Germany, Sweden, France, the UK, Netherlands, and
    Switzerland have high disposable incomes, indicating strong potential for premium
    skincare product sales.
- **Stable Economic Conditions**: ...
- **Consumer Preferences**: ...

### Recommendation:

Entering the skincare markets in Germany, Sweden, France, Italy, the UK, the Netherlands, and
    Switzerland is financially viable and potentially profitable due to their stable economic
    conditions, high purchasing power, and alignment with consumer preferences for premium,
    sustainable, and innovative products.
```

```
logistics_agent
Query:
Evaluate the logistical and supply chain requirements for distributing skincare products
    across Germany and other identified high-demand European markets, focusing on high-demand
    regions.

Response:
To evaluate the logistical and supply chain requirements for distributing skincare products
    across Germany and other identified high-demand European markets, we need to consider the
    infrastructure, regulatory environment, and distribution networks in these regions.

### Logistical and Supply Chain Evaluation:

1. **Germany**:
   - **Infrastructure**: Highly efficient with robust road, rail, and port systems. Central
       location within Europe facilitates access to various markets.
   - **Regulatory Environment**: Supportive of business operations with strong manufacturing
       and distribution networks.
   - **Distribution Networks**: Well-developed, enabling efficient distribution across the
       country and to neighboring regions.
...

### Final Recommendation:

- **Germany**: Leverage its central location and efficient infrastructure for distribution
    across Europe. Its strong regulatory environment supports seamless operations.
...

Overall, these countries offer favorable logistical and supply chain conditions for
    distributing skincare products, aligning with consumer preferences and economic
    indicators. Entering these markets is viable and potentially profitable, given their
    infrastructure, regulatory support, and distribution capabilities.
```

```
decision_agent
Query:
Synthesize the insights from market_agent, finance_agent, and logistics_agent to provide a
    comprehensive recommendation for entering high-demand skincare markets in Europe,
    focusing on Germany and other identified regions.
```

```
Response:
### Comprehensive Recommendation for Entering High-Demand Skincare Markets in Europe

Based on the synthesis of insights from consumer trends, economic indicators, and logistical
    evaluations, the following comprehensive recommendation is provided for entering high-
    demand skincare markets in Europe, focusing on Germany and other identified regions:

#### Step 1: Consumer Trends and Preferences Analysis
- **Germany**: High demand for personalized, eco-friendly, and tech-driven skincare products.
...

#### Step 2: Economic Indicators and Purchasing Power Analysis
...

#### Step 3: Logistical and Supply Chain Evaluation
...

### Final Recommendation:

1. **Market Entry Strategy**:
   - **Germany**: Serve as a central hub for distribution across Europe, leveraging its
       efficient infrastructure and strong regulatory support.
   ...

2. **Product Strategy**: ...

3. **Operational Strategy**: ...

By focusing on these high-demand regions and aligning product offerings with consumer
    preferences and economic conditions, the entry into European skincare markets is
    positioned for success, with potential profitability driven by strong purchasing power
    and stable economic indicators.
```

```
market_kb
input:
consumer data in European skincare market.

output:
**Consumer Data in the European Skincare Market**

**Overview:**
The European skincare market remains one of the largest and most mature sectors within the
    beauty and personal care industry. As of 2023, the market continues to exhibit steady
    growth driven by various factors including increasing consumer awareness of skincare
    ingredients, a rising preference for sustainable and ethical products, and the influence
    of digital platforms.

**Consumer Demographics:**
1. **Age Groups:**
...

**Consumer Preferences:**
1. **Ingredient Consciousness:**
...

**Influence of Digital Platforms:**
...

**Challenges:**
...

**Conclusion:**
The European skincare market is diverse and dynamic, with an increasing focus on products that
    prioritize health, sustainability, and ethical standards. Brands that adapt to these
    changing consumer preferences and leverage digital innovations are well-positioned to
    succeed in this competitive landscape.
```

```
finance_forecaster
input:
Financial modeling and cost projections for distributing skincare products across high-demand
    European markets, considering logistics and supply chain requirements.

output:
To forecast the financial outlook for distributing skincare products across high-demand
    European markets, we need to consider several key elements: market size, demand growth,
    logistics and supply chain costs, pricing strategy, and competitive landscape. Below is a
    simplified projection model.

1. **Revenue Forecast:**
```

```
  - **Market Demand Size:** Assume penetration into markets like Germany, France, and the UK (
      collectively accounting for a large portion of the EU skincare market).
  - **Demand Growth:** Skincare product demand is expected to grow at approximately 5-7%
      annually due to increased consumer awareness and preference for skincare. Given the
      high demand and quality of products, you might capture a market share of 0.5-1% in the
      first year.
  - **Average Retail Price:** Estimate an average retail price for your skincare product at
      EUR 50 per unit.
  - **Units Sold:** Initial distribution could lead to approximately 100,000 units sold across
      all markets in the first year, expanding significantly thereafter.

  **Projected Revenue First Year:**
  Revenue = Units Sold * Average Retail Price
  Revenue = 100,000 * EUR 50 = EUR 5,000,000

2. **Cost Forecast:**
   ...

3. **ROI Analysis:**
   ...

**Forecast Summary:**

If executed well with strong marketing, streamlined supply chain, and effective distribution,
    entering high-demand European markets could achieve substantial revenue growth and
    profitability. Gradual market share increases and scaling operations should provide
    opportunities to optimize costs and improve net profits. Investments in logistics
    optimization, brand establishment, and customer loyalty programs could enhance ROI over
    the subsequent years, potentially reaching 40-50% as operations stabilize and market
    presence strengthens.
```

```
logistics_comparator:
input:
{
  "options": [
    "Germany",
    "France",
    "United Kingdom",
    "Italy",
    "Spain"
  ],
  "criteria": "logistical and supply chain requirements for high-demand regions"
}

output:
When considering 'logistical and supply chain requirements for high-demand regions,' there are
    several factors to take into account, including infrastructure efficiency, existing
    trade networks, regulatory environment, and geographic location. Here's a ranking based
    on these considerations:

1. **Germany**: Germany is known for its highly efficient infrastructure system, including
    roads, railways, and ports. It has central access to the European market, making it a
    pivotal hub for logistics and supply chain operations. Its regulatory framework generally
    supports business operations, and it has strong manufacturing and distribution networks.

2. **United Kingdom**:
...

3. **France**:
...

4. **Italy**:
...

5. **Spain**:
...

This ranking reflects the complexity and efficiency of the logistics and supply chain sector
    in each country, considering their ability to manage and fulfill demands across high-
    demand regions efficiently.
```

```
strategic_computation
input:
{
  "expression": "Calculate the new profit margin after a 5% price increase, assuming costs
      remain constant and demand elasticity is low."
}
```

```
output:
To evaluate the new profit margin after a 5% price increase, we need to understand how the
    increase in price affects revenue and then calculate the resulting profit margin under
    the assumption that costs remain constant and demand elasticity is low.

Let's break down the process:

1. **Initial Setup**:
   - Assume the original selling price is P.
   - Assume the cost per unit is C.
   - Assume the initial quantity sold is Q.

2. **Initial Profit Margin**:
   - Initial revenue: R = P * Q
   - Initial total cost: T = C * Q
   - Initial profit: Profit = R - T = (P * Q) - (C * Q) = (P - C) * Q
   - Initial profit margin: Profit Margin = Profit / R = ((P - C) * Q) / (P * Q) = (P - C) / P

3. **Price Increase**:
   - New selling price: P' = P * 1.05

4. **Revenue with Low Demand Elasticity**:
   - With low demand elasticity, quantity sold Q' ~= Q
   - New revenue: R' = P' * Q' = (P * 1.05) * Q = 1.05 * P * Q

5. **New Profit**:
   - New profit: Profit' = R' - T = 1.05 * P * Q - C * Q = (1.05P - C) * Q

6. **New Profit Margin**:
   - Profit Margin' = Profit' / R' = ((1.05P - C) * Q) / (1.05 * P * Q) = (1.05P - C) / (1.05P)

Thus, the new profit margin after a 5% price increase with constant costs and low demand
    elasticity is:

   Profit Margin' = (1.05P - C) / (1.05P)

Compared to the initial profit margin of (P - C) / P, this indicates an increased profit
    margin, assuming demand remains largely unchanged.
```

User-to-agent and agent-to-resource relationships are encoded in a hierarchical access graph, where edges denote authorized delegation. This graph is intentionally non-fully connected to reflect realistic enterprise constraints. Notably, only the *strategy director* user has comprehensive visibility and access across all agents and resources, enabling them to perform holistic decision-making and synthesize the final output. Details of the access graph structure can be found in Section D.2.1, and a full description of the agents and resources is provided in Section D.2.2.

During the experiments, low-level users—assigned to *market researcher*, *financial analyst* and *logistics and operations lead*—are each responsible for a subtask decomposed from the main query. The *strategy director*, acting as the decision maker, is directly assigned the main query and approaches it by following the subtasks from the perspective of each specialized role, synthesizing the final response accordingly.

```
You are a project coordinator tasked with breaking down the following high-level business
    objective into specific subtasks for a team of three users with distinct functional roles
    .

Project: <query>

The users and their roles are:
- user_1: market
- user_2: finance
- user_3: logistics

For each user, generate a role-specific subtask they would be responsible for in this project.
    Output a valid JSON object mapping each user ID to their subtask. Only output the JSON,
    in the following format:

{
  "user_1": "Subtask for market role...",
  "user_2": "Subtask for finance role...",
  "user_3": "Subtask for logistics role...",
}
```

### D.2.1 ACCESS GRAPH CONFIGURATION

The access graph in our system consists of two layers of asymmetric permissions: (1) **user-to-agent** access and (2) **agent-to-resource** access. Each user can only interact with a subset of agents, and each agent can only retrieve a subset of external resources. This layered structure enforces partial observability and decentralized delegation.

Table 6: User-to-agent mapping indicating which agents are accessible to each user role in the asymmetric collaborative scenario.

| User Role | Accessible Agents |
|---|---|
| Market Researcher | `market_agent` |
| Financial Analyst | `finance_agent,` `decision_agent` |
| Logistics Lead | `logistics_agent,` `finance_agent` |
| Strategy Director | `market_agent, finance_agent,` `logistics_agent, decision_agent` |

Table 7: Mapping of agents to their accessible resources in the asymmetric collaborative scenario.

| Agent | Accessible Resources |
|---|---|
| `market_agent` | `market_kb,` `strategic_computation` |
| `finance_agent` | `finance_forecaster,` `strategic_computation` |
| `logistics_agent` | `logistics_comparator,` `strategic_computation` |
| `decision_agent` | `market_kb, finance_forecaster,` `logistics_comparator, strategic_computation` |

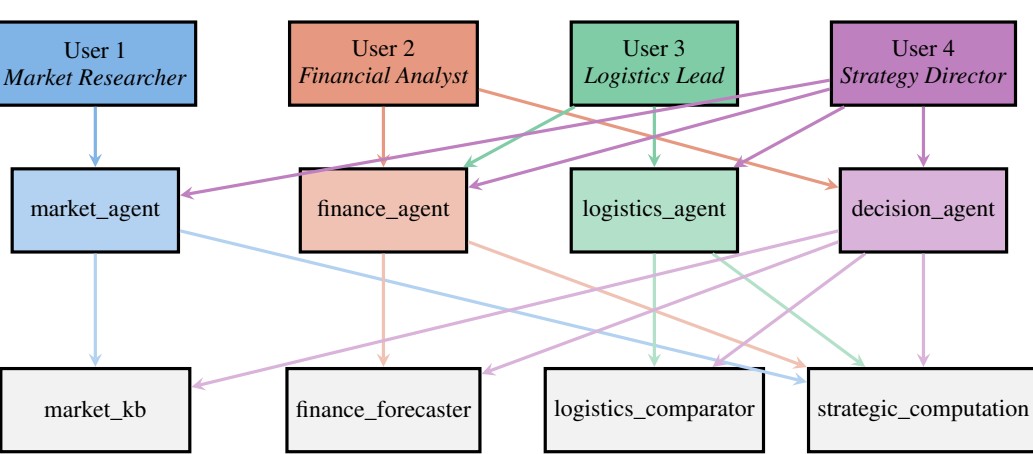

Figure 10: Color-coded access graph illustrating hierarchical user-to-agent and agent-to-resource mappings. Each user and their corresponding agents are assigned a unique color. The *strategy director* (User 4) has full visibility and access.

### D.2.2 AGENT/KNOWLEDGE BASE CONFIGURATION

This section provides detailed configurations of the domain-specific agents, the coordinator and aggregator roles, and the resources they rely on. Each agent is designed with a specialized system prompt tailored to its functional expertise, enabling focused task execution within collaborative workflows. The coordinator assigns queries to appropriate agents based on domain relevance, while the aggregator synthesizes outputs into a coherent final response. Additionally, a set of task-relevant resources—including knowledge bases, forecasting tools, and computation modules—support the agents in delivering accurate and context-aware outputs. Tables 8–10 summarize the roles, descriptions, and system prompts for all components.

Table 8: Domain agents and their system prompts in the asymmetric collaborative scenario.

| Agent | Description | System Prompt |
|---|---|---|
| market_agent | Analyzes market competition and consumer data. | You are a market intelligence agent. Use market research reports and strategic case studies to identify trends, consumer behaviors, and competition analysis. Always consult memory first. If needed, call tools to retrieve new insights. |
| finance_agent | Performs financial modeling and cost projections. | You are a financial planning agent. Evaluate revenue forecasts, pricing structures, and cost breakdowns using financial models and simulations. |
| logistics_agent | Evaluates supply chain feasibility and vendor performance. | You are a logistics analyst. Use distribution comparators and operational data to assess fulfillment feasibility and vendor efficiency. |
| decision_agent | Synthesizes recommendations from other domains. | You are a strategic coordinator. Integrate inputs from various domain experts to formulate a cohesive strategic recommendation. |

Table 9: Coordinator and aggregator with their system prompts in the asymmetric collaborative scenario.

| | Description | System Prompt |
|---|---|---|
| coordinator | Assigns queries to appropriate agents based on domain expertise. | You are a coordinator for a specialized multi-agent system. Your job is to analyze user queries and determine which specialized agent is best suited to handle each query or sub-query. For each query: 1. Identify the primary domain(s) of the query 2. Select the most appropriate specialized agent based on their expertise areas. 3. For complex queries that span multiple domains, determine if sequential agent consultation is needed 4. When reformulating queries for specific agents, emphasize aspects relevant to their expertise 5. When the task is complete, respond with COMPLETE |
| aggregator | Merges multi-agent outputs into a unified user response. | You are an aggregator for a multi-agent system. Combine outputs from multiple specialized agents into a single, direct, and coherent response to the user query. |

Table 10: Available resources and their functional roles in the asymmetric collaborative scenario.

| Name | Type | Description |
|------|------|-------------|
| market_kb | knowledge_base | Market trends and consumer data |
| finance_forecaster | forecast_model | Performs financial forecasting and ROI estimation |
| logistics_comparator | comparator | Compares distribution and vendor options for supply chain planning |
| strategic_computation | computation | Executes strategic cost-benefit calculations |

## D.3 EXAMPLES

### D.3.1 READ/WRITE EXAMPLE

Table 11 presents a multi-agent collaboration example in which agents write to and read from both private (user-specific) and public (cross-user) memory. This scenario highlights how agents can leverage memory entries for continued reasoning and strategic synthesis across users and roles.

Table 11: Memory write–read examples for multi-user multi agent collaboration, showing both user-specific and cross-user entries.

*project_hard_52: "Craft a strategic growth plan for smart home devices in privacy-regulated markets."*

**Private Memory**
*Discussion: This fragment is utilized in the second* market_agent *entry to inform the financial modeling of strategic initiatives focused on privacy-regulated markets.*

| | |
|---|---|
| **Memory ID** | 51982b4d-3fc7-4dc8-83b7-e34ca07114ca |
| **Key** | Growth Opportunities in Privacy-Regulated Markets |
| **Value** | Developing new products with advanced privacy features can capture market interest. Targeting regions with stringent **privacy** regulations can open new markets. Consumer education campaigns can differentiate a brand and build trust. |
| **Generated at** | market_agent (user_4): *Conduct market analysis to identify consumer preferences and trends in privacy-regulated markets, focusing on smart home devices. Assess competitive landscape and potential growth opportunities.* |
| **Used at** | market_agent (user_4): *Develop a financial model to evaluate the cost-effectiveness and ROI of strategic initiatives in **privacy-regulated** markets, considering strict resource constraints. Use insights from the market analysis on consumer preferences, trends, and competitive landscape to inform the financial projections.* |

**Shared Memory**
*Discussion: This is leveraged in the second* decision_agent *of **user_4** entry to synthesize insights from the market analysis and financial modeling when crafting a growth plan for smart home devices in privacy-regulated markets under strict resource constraints*

| | |
|---|---|
| **Memory ID** | 54214a00-8806-4ada-98ae-7f4029c06b90 |
| **Key** | strategic recommendations for privacy-regulated markets |

| | |
|---|---|
| **Value** | 1. **Optimize Initial Investment** by reducing upfront costs through leveraging existing technologies, partnerships, or phased implementation. 2. **Reduce Operating Costs** by outsourcing non-core activities, automating compliance processes, and negotiating better rates with vendors. 3. **Accelerate Time to Market** using agile project management, early regulatory engagement, and rapid prototyping. 4. **Enhance Market Penetration** through targeted marketing, leveraging customer insights, and building strategic alliances. 5. **Adjust Revenue Growth Expectations** by identifying additional revenue streams and expanding into adjacent markets. 6. **Reevaluate Discount Rate** by implementing risk mitigation strategies like securing insurance and diversifying investments. 7. **Scenario Planning** for best-case, worst-case, and most likely outcomes to understand variability in returns. |
| **Generated at** | `decision_agent` (user_2): *Synthesize recommendations based on the financial model developed by finance_agent, focusing on strategic initiatives in privacy-regulated markets with strict resource constraints.* |
| **Used at** | `decision_agent` (user_4): *Synthesize the insights from the **market analysis** and **financial modeling** to craft a strategic growth plan for smart home devices in privacy-regulated markets. Consider the strict resource constraints and cross-domain dependencies, and provide a comprehensive final recommendation.* |

### D.3.2 WITHOUT SUBTASK DECOMPOSITION

Table 12 shows that without subtask decomposition, limited agent and resource access can prevent effective query handling. When the query is reformulated to match User 3's operational role, the system produces a detailed response. In contrast, the original high-level query fails due to misalignment with agent capabilities. These results demonstrate the system's ability to address complex real-world tasks via coordinated collaboration among role-specialized users, agents, and resources.

Table 12: Comparison of user's responses with and without subtask decomposition.

| | |
|---|---|
| *project_easy_01: "Identify high-demand skincare markets in Europe."* | |
| **With Subtask (User 3)** | |
| **Query** | Evaluate the logistical and supply chain requirements for distributing skincare products across different European markets, focusing on high-demand regions. |
| **Response** | To conduct a comprehensive analysis of the logistical and supply chain requirements for distributing skincare products across high-demand European markets, we need to consider both logistical infrastructure and financial projections. **Logistical Analysis:** 1. **Germany**: Germany's central location and efficient infrastructure make it an ideal hub for distribution. Its strong manufacturing and distribution networks support seamless logistics operations, providing easy access to various European markets. 2. **United Kingdom**: ... |
| **Without Subtask (User 3)** | |
| **Query** | Identify high-demand skincare markets in Europe. |
| **Response** | No agents were able to process the query. |

### D.4 POLICY GRANULARITY OVERHEAD ANALYSIS

We quantify the computational overhead associated with increasingly fine-grained transformation policies to assess the practical scalability of sophisticated access control mechanisms. This analysis systematically measures LLM invocation frequency across policy complexity levels, providing empirical evidence for the feasibility of multi-tiered policy enforcement in production environments.

We evaluate three policy granularity configurations representing different levels of access control sophistication:

- **System-level policies:** Global transformation rules applied uniformly across all interactions

- **System + Agent-level policies:** Per-agent specialization overlaid on system-wide constraints

- **System + Agent + User-level policies:** Full hierarchical policy enforcement with user-specific customization

Table 13 presents the mean LLM call frequency per user role across policy configurations. Results indicate that policy granularity introduces minimal computational overhead, with variations in total LLM calls remaining within statistical noise bounds. Notably, even maximum granularity configurations show negligible increases relative to baseline system-level policies, confirming that sophisticated multi-tier access control mechanisms maintain computational efficiency while providing enhanced security guarantees.

Table 13: Total LLM calls by policy granularity across different user roles.

| User Role | System + Agent + User | System + Agent | System Only |
|---|---|---|---|
| Market Researcher | $3.77 \pm 4.84$ | $3.84 \pm 4.69$ | $4.23 \pm 4.92$ |
| Financial Analyst | $11.81 \pm 9.31$ | $12.69 \pm 9.52$ | $10.09 \pm 5.93$ |
| Logistics & Ops Lead | $12.93 \pm 8.41$ | $12.08 \pm 7.64$ | $12.12 \pm 8.81$ |
| Strategy Director | $24.75 \pm 12.83$ | $25.49 \pm 12.54$ | $24.09 \pm 13.68$ |

# E  SCENARIO 3: DYNAMICALLY EVOLVING COLLABORATIVE MEMORY

## E.1  DATASET DETAILS

The SciQAG dataset covers 24 Web of Science (WoS) categories. Since our experiments target evaluating system behavior under dynamic access settings without requiring the full dataset, we select 5 categories: "*Chemistry, Analytical*", "*Energy & Fuels*", "*Materials Science, Paper & Wood*", "*Materials Science, Ceramics*", and "*Physics, Mathematical*". These five categories cover a broad range of scientific domains. Each instance in this dataset consists of passages from a scientific paper along with corresponding question-answer pairs. The passages serve as the reference corpus for answering the associated questions. Table 14 reports the detailed breakdown of document counts, average lengths, and query-type distributions.

Table 14: Statistics of the subset of SCIQAG dataset for the dynamically evolving collaborative scenario.

| Statistic | Count | Percentage |
|---|---|---|
| **Documents** | | |
| Total articles | 100 | — |
| Average tokens per article | 3,609 | — |
| **Queries (N = 100)** | | |
| Queries from "*Chemistry, Analytical*" | 20 | 20% |
| Queries from "*Energy & Fuels*" | 20 | 20% |
| Queries from "*Materials Science, Paper & Wood*" | 20 | 20% |
| Queries from "*Materials Science, Ceramics*" | 20 | 20% |
| Queries from "*Physics, Mathematical*" | 20 | 20% |

## E.2  EXPERIMENTAL DETAILS

- **Users**: We synthesize queries for 5 users. Each user has 4 randomly sampled queries from each of the 5 categories, resulting in a total of 100 queries.

- **Agents**: There are 5 agents, each specializing in one of the five scientific categories, corresponding to their respective domain expertise.

- **Resources**: There are 5 resources, each consisting of the passages from the dataset associated with a specific category, used to answer the corresponding questions.

We evaluate the framework under varying access graph configurations while accumulating memory across the dynamic experiment. The Shared Memory configuration is enabled to support cross-user memory sharing. The same queries are reused in different graph configurations. Memory retrieval is configured with a user-specific tier size of $k_{\text{user}} = 10$, a shared tier size of $k_{\text{cross}} = 10$, and a similarity threshold of 0.1. The agent has the access to one corresponding resource. We next provide details on the access graph configuration and the agent/knowledge base configurations.

### E.2.1  ACCESS GRAPH CONFIGURATION

We simulate the evolution of the access graph as a stochastic process. For simplicity, we maintain the agent-resource connection as a one-to-one mapping as reported in Table 17 and change the connections between users and agents. Specifically, we gradually add unseen edges to represent the granting of access and remove existing edges to represent the revocation of access. During the granting phase (from $t_0$ to $t_4$), the number of edges increases progressively to 5, 10, 15, 20, and 25, respectively. Starting from the fully connected graph at $t_4$, we reverse the process by progressively removing edges, resulting in graphs with 20, 15, 10, and 5 edges at $t_5$ through $t_8$, respectively. The graph at the previous stage provides the base of the next stage. The pseudocodes for these two procedures are in Algorithm 1 and Algorithm 2. We use $p = 0.2$ for the probability.

---

**Algorithm 1** Access Granting Procedure (Generating $t_0$–$t_4$)

---

1: Initialize empty bipartite graph $G$ with user and agent nodes
2: Define candidate edges between all user-agent pairs
3: **for** $i = 0$ to 4 **do**
4:     Add 5 new edges to $G$ using Bernoulli trials with probability $p$
5:     Collect updated user-to-agent mappings from $G$
6:     Save updated YAML configuration with timestamp $t_i$
7: **end for**

---

**Algorithm 2** Access Revoking Procedure (Generating $t_5$–$t_8$)

---

1: Load fully connected bipartite graph $G$
2: **for** $i = 5$ to 8 **do**
3:     Remove 5 edges from $G$ using Bernoulli trials with probability $p$
4:     Collect updated user-to-agent mappings from $G$
5:     Save updated YAML configuration with timestamp $t_i$
6: **end for**

---

The user-to-agent mappings at each timestamp during the granting process are summarized in Table 15, and the mappings for the revoking process are shown in Table 16.

Table 15: Users to their accessible agents at timestamp from $t_0$ to $t_4$ in the access granting stage.

| Timestamp | User | Accessible Agents |
|---|---|---|
| $t_0$ | $U_1$ | `materials_paper_wood_agent,materials_ceramics_agent` |
| | $U_2$ | `physics_mathematical_agent` |
| | $U_3$ | |
| | $U_4$ | `materials_paper_wood_agent` |
| | $U_5$ | `chemistry_analytical_agent` |
| $t_1$ | $U_1$ | `materials_paper_wood_agent,materials_ceramics_agent,` `energy_fuels_agent` |
| | $U_2$ | `physics_mathematical_agent,energy_fuels_agent` |
| | $U_3$ | `energy_fuels_agent,chemistry_analytical_agent` |
| | $U_4$ | `materials_paper_wood_agent,energy_fuels_agent` |
| | $U_5$ | `chemistry_analytical_agent` |
| $t_2$ | $U_1$ | `materials_paper_wood_agent,materials_ceramics_agent,` `energy_fuels_agent` |
| | $U_2$ | `physics_mathematical_agent,energy_fuels_agent` `chemistry_analytical_agent` |
| | $U_3$ | `energy_fuels_agent,chemistry_analytical_agent,` `materials_ceramics_agent` |
| | $U_4$ | `materials_paper_wood_agent,energy_fuels_agent,` `physics_mathematical_agent` |
| | $U_5$ | `chemistry_analytical_agent,materials_ceramics_agent,` `energy_fuels_agent` |
| $t_3$ | $U_1$ | `materials_paper_wood_agent,materials_ceramics_agent,` `energy_fuels_agent,chemistry_analytical_agent` |
| | $U_2$ | `physics_mathematical_agent,energy_fuels_agent,` `chemistry_analytical_agent,materials_ceramics_agent` |
| | $U_3$ | `energy_fuels_agent,chemistry_analytical_agent,` `materials_ceramics_agent,physics_mathematical_agent` |
| | $U_4$ | `materials_paper_wood_agent,energy_fuels_agent,` `physics_mathematical_agent,materials_ceramics_agent` |
| | $U_5$ | `chemistry_analytical_agent,materials_ceramics_agent,` `energy_fuels_agent,physics_mathematical_agent` |
| $t_4$ | $U_1$ | `materials_paper_wood_agent,materials_ceramics_agent,` `energy_fuels_agent,chemistry_analytical_agent,` `physics_mathematical_agent` |
| | $U_2$ | `physics_mathematical_agent,energy_fuels_agent,` `chemistry_analytical_agent,materials_ceramics_agent,` `materials_paper_wood_agent` |
| | $U_3$ | `energy_fuels_agent,chemistry_analytical_agent,` `materials_ceramics_agent,physics_mathematical_agent,` `materials_paper_wood_agent` |
| | $U_4$ | `materials_paper_wood_agent,energy_fuels_agent,` `physics_mathematical_agent,materials_ceramics_agent,` `chemistry_analytical_agent` |
| | $U_5$ | `chemistry_analytical_agent,materials_ceramics_agent,` `energy_fuels_agent,physics_mathematical_agent,` `materials_paper_wood_agent` |

Table 16: Users to their accessible agents at timestamp from $t_5$ to $t_8$ in the access revoking stage.

| Timestamp | User | Accessible Agents |
|---|---|---|
| $t_5$ | $U_1$ | chemistry_analytical_agent, energy_fuels_agent, physics_mathematical_agent |
| | $U_2$ | chemistry_analytical_agent, energy_fuels_agent, materials_paper_wood_agent, materials_ceramics_agent |
| | $U_3$ | chemistry_analytical_agent, energy_fuels_agent, materials_paper_wood_agent, materials_ceramics_agent, physics_mathematical_agent |
| | $U_4$ | chemistry_analytical_agent, energy_fuels_agent, materials_ceramics_agent, physics_mathematical_agent |
| | $U_5$ | energy_fuels_agent, materials_paper_wood_agent, materials_ceramics_agent, physics_mathematical_agent |
| $t_6$ | $U_1$ | energy_fuels_agent |
| | $U_2$ | chemistry_analytical_agent, energy_fuels_agent, materials_ceramics_agent |
| | $U_3$ | chemistry_analytical_agent, materials_paper_wood_agent, materials_ceramics_agent, physics_mathematical_agent |
| | $U_4$ | chemistry_analytical_agent, energy_fuels_agent, materials_ceramics_agent, physics_mathematical_agent |
| | $U_5$ | energy_fuels_agent, materials_paper_wood_agent, physics_mathematical_agent |
| $t_7$ | $U_1$ | |
| | $U_2$ | chemistry_analytical_agent, energy_fuels_agent, materials_ceramics_agent |
| | $U_3$ | materials_paper_wood_agent, materials_ceramics_agent |
| | $U_4$ | materials_ceramics_agent, physics_mathematical_agent |
| | $U_5$ | energy_fuels_agent, materials_paper_wood_agent, physics_mathematical_agent |
| $t_8$ | $U_1$ | |
| | $U_2$ | chemistry_analytical_agent |
| | $U_3$ | materials_ceramics_agent |
| | $U_4$ | materials_ceramics_agent |
| | $U_5$ | energy_fuels_agent, physics_mathematical_agent |

Table 17: Agents to their accessible resources in the dynamically evolving collaborative scenario.

| Agent | Accessible Resources |
|---|---|
| materials_paper_wood_agent | materials_paper_wood_kb |
| materials_ceramics_agent | materials_ceramics_kb |
| energy_fuels_agent | energy_fuels_kb |
| chemistry_analytical_agent | chemistry_analytical_kb |
| physics_mathematical_agent | physics_mathematical_kb |

### E.2.2 AGENT/KNOWLEDGE BASE CONFIGURATION

We deploy five specialist agents, each paired with a dedicated knowledge base. Table 18 list each agent's description and full system prompt. In addition to the specialist agents, there is a coordinator to assign queries to different agents and an aggregator to merge the outputs. Their system prompts are listed in Table 19.

Table 18: Domain agents and their system prompts in the dynamically evolving collaborative scenario.

| Agent | Description | System Prompt |
|---|---|---|
| chemistry_ analytical_ agent | Specialist in analytical chemistry—techniques, instrumentation, and chemical analysis. | You are an Analytical Chemistry specialist with deep expertise in:
• Explaining analytical methods such as spectroscopy, chromatography, and mass spectrometry
• Interpreting experimental results and chemical measurements
• Discussing laboratory techniques and instrumentation details
• Providing insights on quality control and chemical validation processes
Always check relevant memories first. When those are insufficient, use the knowledge_base tools. Prioritize information from memories and the scientific literature dataset over your parametric knowledge. Do not provide responses to inquiries that are unrelated to the domain of analytical chemistry. |
| energy_ fuels_ agent | Specialist in energy and fuels—renewable energy, fossil fuels, and energy technologies. | You are an Energy & Fuels specialist with deep expertise in:
• Discussing renewable energy sources like solar, wind, and bioenergy
• Analyzing fossil fuel technologies including oil, gas, and coal
• Evaluating energy storage systems and grid technologies
• Interpreting policies, market trends, and environmental impacts
Always check relevant memories first. When those are insufficient, use the knowledge_base tools. Prioritize information from memories and the energy sector dataset over your parametric knowledge. Do not provide responses to inquiries that are unrelated to the domain of energy and fuels. |

Table 18: Domain agents and their system prompts in the dynamically evolving collaborative scenario. (continued)

| Agent | Description | System Prompt |
|---|---|---|
| `materials_ paper_ wood_ agent` | Specialist in materials science—paper, wood, and biomaterials research and applications. | You are a Paper & Wood Materials Science specialist with deep expertise in:
• Explaining the properties and processing of paper and wood materials
• Discussing advancements in sustainable materials and bio-based composites
• Analyzing mechanical, chemical, and environmental performance
• Exploring industrial applications in packaging, construction, and manufacturing
Always check relevant memories first. When those are insufficient, use the knowledge_base tools. Prioritize information from memories and the materials science dataset over your parametric knowledge. Do not provide responses to inquiries that are unrelated to the domain of paper and wood materials. |
| `materials_ ceramics_ agent` | Specialist in materials science—ceramics, glass, and high-performance materials. | You are a Ceramics Materials Science specialist with deep expertise in:
• Explaining the processing and properties of ceramics and glass
• Analyzing thermal, mechanical, and electrical performance characteristics
• Discussing applications in aerospace, electronics, and structural components
• Exploring advances in high-temperature and wear-resistant materials
Always check relevant memories first. When those are insufficient, use the knowledge_base tools. Prioritize information from memories and the materials science dataset over your parametric knowledge. Do not provide responses to inquiries that are unrelated to the domain of ceramics and high-performance materials. |
| `physics_ mathematical_ agent` | Specialist in mathematical physics—models, theories, and mathematical formulations of physical systems. | You are a Mathematical Physics specialist with deep expertise in:
• Explaining mathematical models of physical systems and phenomena
• Interpreting equations, simulations, and theoretical predictions
• Discussing areas such as quantum mechanics, statistical physics, and dynamical systems
• Providing insights into mathematical methods used in physics research
Always check relevant memories first. When those are insufficient, use the knowledge_base tools. Prioritize information from memories and the physics literature dataset over your parametric knowledge. Do not provide responses to inquiries that are unrelated to the domain of mathematical physics. |

Table 19: Coordinator and aggregator with their system prompts in the dynamically evolving collaborative scenario.

| | Description | System Prompt |
|---|---|---|
| coordinator | Assigns queries to appropriate agents based on domain expertise. | You are a coordinator for a specialized scientific knowledge system consisting of domain-specific expert agents. Your job is to analyze user queries and determine which specialized agent is best suited to handle each query or sub-query. 

 For each query: 
 1. Identify the primary domain(s) of the query 
 2. Select the most appropriate specialized agent based on their expertise areas. 
 3. For complex queries that span multiple domains, determine if sequential agent consultation is needed 
 4. When reformulating queries for specific agents, emphasize aspects relevant to their expertise 
 5. When the task is complete, respond with COMPLETE |
| aggregator | Merges multi-agent outputs into a unified user response. | You are an aggregator for a multi-agent scientific knowledge system. Your role is to combine outputs from multiple specialized agents into a single, logically structured, and detailed response to the user query. |

### E.3 EXAMPLES

To illustrate how our system stores and subsequently reuses memory entries and how the saved memory facilitates solving subsequent queries, Table 20 presents two representative "write–then–read" cases using private and shared memory fragments. Each example includes the memory's unique ID, the stored key–value pair, the original query that triggered the memory write, and the later query that retrieved it. Example 1 demonstrates private memory reuse: the memory was created when user_1 submitted Query #18 and later accessed by the same user in Query #40. Example 2 highlights shared memory reuse: information originally stored in response to user_5's query was subsequently retrieved by user_2. We provide detailed discussions in the table.

### E.4 RAW PERFORMANCE DATA AND COMPLETE ACCESS MATRIX UNDER DIFFERENT ACCESS GRAPH CONFIGURATIONS

In this section, we present the results across graph configurations, including (1) raw data of the system performance over eight time blocks with dynamically changing privileges and (2) the full access matrix in Figure 11.

Figure 11 presents the complete agent and resource usage patterns across user queries. It clearly illustrates that agents and resources are only utilized when access is explicitly granted (i.e., in the yellow rectangles) despite the change of the privileges.

### E.5 SYSTEM SCALABILITY ANALYSIS

We evaluate the computational scalability of our framework under increased system complexity to establish its viability for enterprise-scale deployments. This analysis systematically stress-tests performance by scaling the user-agent access graph from 5 users with 10 permission edges to 50 users with 100 edges, representing a 10× increase in both user population and access relationship complexity.

Table 22 reports key performance metrics across graph configurations while maintaining a fixed access sparsity of 0.60. Results demonstrate graceful scaling behavior: despite the substantial increase

Table 20: Memory write–read examples for private and shared memories in the dynamically evolving collaborative scenario.

**Example 1 (Private Memory)**

*Discussion: The stored content discusses structural dimorphism in cellulose, emphasizing the role of 21 symmetry structures as high-energy intermediates, thereby supporting analysis of flexibility in cellulose's structural transitions.*

| | |
|---|---|
| **Memory ID** | 8177e2aa-2ef0-4b20-bdcb-af33d8d1ae00 |
| **Key** | Potential energy surfaces for cellulose |
| **Value** | The study concluded that twofold symmetry structures of cellulose act as barriers between lower-energy forms. Energy maps using empirical force fields and quantum mechanical methods suggested that 21 symmetry structures might have higher potential energy due to close atomic contacts, potentially serving as transition points or barriers. Despite the increased energy from these close contacts, hydrogen bonds could stabilize these structures. The study also highlighted the flexibility of cellulose chains and possible deviations from 21 symmetry, indicating substantial flexibility and the influence of crystal packing on the flat ribbon-like 21 structures. |
| **Generated at** | Query #18 (user_1): *What was the conclusion about the potential energy surfaces for cellulose and its derivatives?* |
| **Used at** | Query #40 (user_1): *What is the significance of cellulose's dimorphism?* |

**Example 2 (Shared Memory)**

*Discussion: Although the original context involved irradiated wood, the saved explanation—how moisture accelerates hydrolytic and chemical degradation—proved relevant to understanding degradation processes in insulating paper. Additionally, this sharing occurs because both User_2 and User_5 have access to the chemistry analytical agent. The constraints enforced by our access graph ensure that the shared information is used safely and appropriately.*

| | |
|---|---|
| **Memory ID** | 01b1f4d0-d7d5-4be5-ae96-98620a49add2 |
| **Key** | Influence of relative humidity on irradiated wood |
| **Value** | Relative humidity affects the chemical changes in irradiated wood during heat treatment, influencing color change. Higher humidity increases moisture content, facilitating hydrolytic reactions and lignin breakdown, leading to chromophore formation and color change. Moisture acts as a plasticizer, enhancing polymer mobility and reaction rates, which accelerates lignin breakdown. Controlling humidity can manage color alteration in wood. |
| **Generated at** | Query #21 (user_5): *How does relative humidity affect the change in color of irradiated wood during heat treatment?* |
| **Used at** | Query #66 (user_2): *What role does moisture play in the degradation of insulating papers?* |

Table 21: Performance and Memory Statistics Across Different Access Graph Configurations

| Time | Condition | Accuracy | # Resources | # Agent | # User Memory | # Cross Memory |
|---|---|---|---|---|---|---|
| $t_0$ | Grant | $0.27 \pm 0.01$ | $0.46 \pm 0.02$ | $0.51 \pm 0.03$ | $2.94 \pm 0.93$ | $3.25 \pm 0.70$ |
| $t_1$ | Grant | $0.40 \pm 0.02$ | $0.34 \pm 0.02$ | $0.74 \pm 0.04$ | $5.85 \pm 0.14$ | $7.09 \pm 0.37$ |
| $t_2$ | Grant | $0.54 \pm 0.03$ | $0.60 \pm 0.03$ | $1.14 \pm 0.06$ | $9.81 \pm 0.57$ | $11.40 \pm 0.91$ |
| $t_3$ | Grant | $0.56 \pm 0.03$ | $0.57 \pm 0.03$ | $1.29 \pm 0.06$ | $11.11 \pm 1.04$ | $12.90 \pm 0.83$ |
| $t_4$ | Grant | $0.61 \pm 0.03$ | $0.48 \pm 0.02$ | $1.31 \pm 0.07$ | $11.27 \pm 0.60$ | $13.10 \pm 0.43$ |
| $t_5$ | Revoke | $0.56 \pm 0.03$ | $0.21 \pm 0.01$ | $1.19 \pm 0.06$ | $11.57 \pm 0.72$ | $11.90 \pm 0.62$ |
| $t_6$ | Revoke | $0.47 \pm 0.02$ | $0.30 \pm 0.01$ | $1.08 \pm 0.05$ | $10.77 \pm 2.06$ | $10.80 \pm 2.07$ |
| $t_7$ | Revoke | $0.49 \pm 0.02$ | $0.23 \pm 0.01$ | $1.07 \pm 0.05$ | $10.75 \pm 1.44$ | $10.75 \pm 1.44$ |
| $t_8$ | Revoke | $0.37 \pm 0.02$ | $0.06 \pm 0.00$ | $0.68 \pm 0.03$ | $6.75 \pm 0.48$ | $6.75 \pm 0.48$ |

in system complexity, end-to-end execution time increases by only ~10%. Task accuracy remains stable, while resource utilization metrics scale proportionally with increased user demand.

### E.6 COMPARISON WITH MEMORY SHARING BASELINE

We establish the empirical necessity of formal access control through direct comparison with existing memory sharing approaches that lack user-level privacy constraints. This analysis compares our

Table 22: System performance across varying graph sizes (Sparsity = 0.60).

| Configuration | Accuracy | Resource Usage | Agent Usage | Exec. Time (s) |
|---|---|---|---|---|
| 5 users - 5 agents - 10 edges | $0.396 \pm 0.040$ | $0.620 \pm 0.058$ | $0.720 \pm 0.055$ | $14.366 \pm 1.548$ |
| 50 users - 5 agents - 100 edges | $0.421 \pm 0.017$ | $0.814 \pm 0.034$ | $0.866 \pm 0.029$ | $15.863 \pm 0.519$ |
| **Relative Change** | +6.3% | +31.3% | +20.3% | +10.4% |

framework against the Memory Sharing baseline Gao & Zhang (2024), which implements unrestricted memory pooling across agents without consideration of multi-user access policies. The comparison elucidates the fundamental security-utility trade-offs inherent in collaborative memory systems.

Table 23 reports performance metrics under controlled experimental conditions (5 users, 5 agents, 10 access edges, sparsity = 0.60). While both approaches achieve comparable task accuracy and computational efficiency, they exhibit fundamentally different security characteristics. The baseline Memory Sharing approach demonstrates a critical vulnerability: 59.03% information leakage ratio, indicating frequent exposure of memory fragments to unauthorized users. Conversely, our Collaborative Memory framework achieves zero information leakage while preserving task performance, representing complete policy compliance across all access control constraints.

Table 23: Comparison between CollabMem and Memory Sharing baseline (5 users – 5 agents – 10 access edges, Sparsity = 0.60).

| Method | Accuracy | Resource Usage | Agent Usage | Private Mem | Shared Mem | Leakage Ratio |
|---|---|---|---|---|---|---|
| CollabMem | $0.401 \pm 0.020$ | $0.70 \pm 0.035$ | $0.74 \pm 0.037$ | $3.80 \pm 0.490$ | $5.57 \pm 0.369$ | $0.00 \pm 0.00$ |
| MemorySharing | $0.404 \pm 0.020$ | $0.74 \pm 0.037$ | $0.73 \pm 0.036$ | N/A | $6.98 \pm 0.485$ | $59.03\% \pm 1.86\%$ |

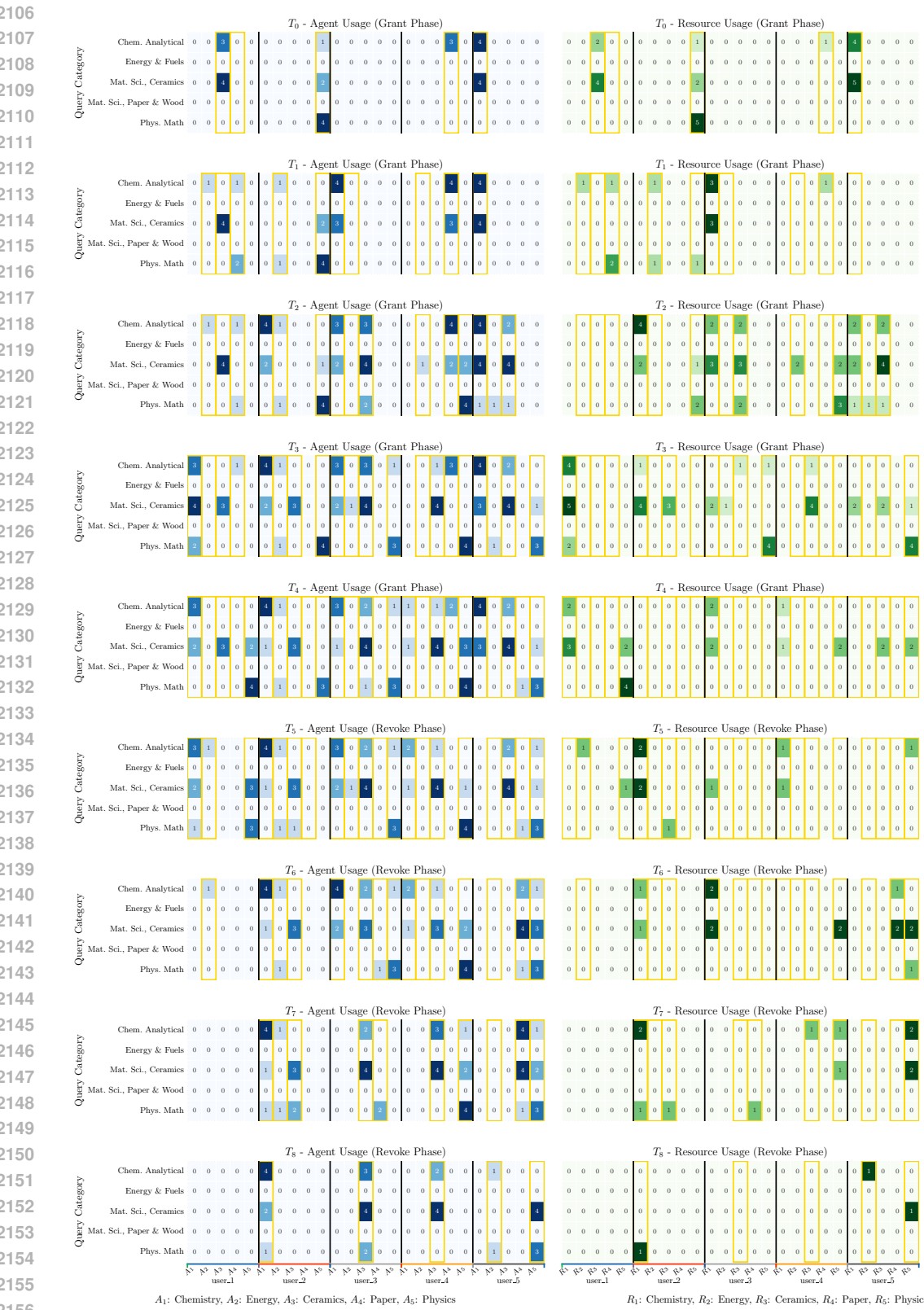

Figure 11: Complete agent and resource usage across user queries from different categories. Yellow rectangles indicate granted access, with values representing the corresponding usage counts.

