# OpenReview forum: "Collaborative Memory: Multi-User Memory Sharing in LLM Agents with Dynamic Access Control"
_ICLR.cc/2026/Conference — Submitted to ICLR 2026_

### Official Review · Reviewer_9nnX · 2025-10-26

**Soundness:** 3
**Presentation:** 2
**Contribution:** 3
**Rating:** 4
**Confidence:** 4

**Summary:**

This paper introduces Collaborative Memory, a framework for managing memory in multi-user, multi-agent systems with asymmetric and time-evolving access controls. The key innovation is the formalization of dynamic permissions using bipartite graphs that connect users, agents, and resources, combined with a two-tier memory architecture. The system enforces fine-grained read and write policies that respect access constraints while enabling cross-user knowledge sharing.

**Strengths:**

1. The two-tier memory system with provenance tracking provides an elegant solution to the privacy-utility tradeoff.

2. Clear design of three progressively complex scenarios effectively demonstrates different aspects of the framework.

3. Results and analyses show significant resource utilization reductions while maintaining accuracy and zero information leakage.

**Weaknesses:**

**1. Limitations in real-world validation**: All experiments use synthetic datasets or benchmarks, while lacking evaluation with real-world multi-user deployments. This raises questions about practical applicability.

**2. Limitations in policy implementations**: The read/write policies (Table 1) are quite basic, consisting of simple transformations that are applied uniformly. However, this paper doesn't explore more sophisticated policies, such as differential privacy, semantic filtering, context-dependent redaction, etc.

**3. Limitations in privacy analysis**: This paper shows no formal privacy guarantees or threat model. The "leakage ratio" metric is introduced only in comparison with baseline. It also lacks discussion of potential attacks or adversarial scenarios, and it is unclear what "zero leakage" actually means, especially in terms of information privacy.

**4. Limitations in scalability:** This paper only tests 5→50 users with 10% execution time increase, but doesn't address the memory-related problems that this paper focuses on, such as the memory storage growth over time, the query latency with large memory pools, the concurrency handling with many simultaneous users, etc.

**5. Limitations in experiments:** It only compares against the Memory Sharing baseline, lacking baselines such as traditional access control systems, other multi-agent memory architectures, privacy-preserving collaborative learning approaches, etc. It also lacks discussions of failure modes and common patterns that can benefit future improvements.

**Questions:**

My questions are following several aspects mentioned in weakness:

- Can you provide formal privacy guarantees for your framework? What specific threat model does the system defend against, and what real-world scenarios this framework can apply and/or generalize to?
- The current read/write policies are fairly simple. Can current framework and findings generalize to more complex policies? What are the results of your investigation on more complex policies?
- In Equation 3, $\mathcal{M}(u,a,t)$ requires both $\mathcal{A}(m) \subseteq \mathcal{A}(u,t)$ and $\mathcal{R}(m) \subseteq \mathcal{R}(a,t)$. Why is this the right semantics? Can there be cases where partial overlap should allow access?
- How do you ensure the quality of shared memories over time? Can incorrect information from one user pollute the shared memory for others?
- How would your framework compare against other baselines, such as traditional access control systems, other multi-agent memory architectures, privacy-preserving collaborative learning approaches, etc.?
- If shared memory contains conflicting information contributed by different users with different access levels, how does the system resolve this during retrieval?

---

> ### Author Response · Authors · 2025-11-26
> **Response to Reviewer 9nnX (1/2)**
>
> Thank you for your thoughtful feedback and insightful questions.
>
> ### Q1. Can you provide formal privacy guarantees? What specific threat model do you consider, and what does “zero leakage” mean?
>
> **A1.**
> We focus on a **cross-user access-control threat model**:
>
> - Users and agents follow the protocol but may attempt to infer information from memory that they are not authorized to see.
> - The underlying infrastructure (storage, LLM provider) is trusted; we do not address lower-level side-channels or fully adversarial code.
>
> Under this model, we provide a *formal access-control guarantee*:
>
> > A fragment is eligible for user \(u\) via agent \(a\) at time \(t\) only if its provenance is consistent with the user’s current permissions (via the user→agent and agent→resource access graphs).
>
> “**Zero leakage**” in our experiments means:
>
> - In the dynamic-permissions Scenario 3, across all queries and time steps, *no* returned fragment’s provenance violated the current access graphs for the querying user.
> - In contrast, the Memory Sharing baseline (single global pool, no per-user access control) exposed unauthorized fragments in roughly 59% of retrievals (as reported in **Appendix E.4, Table 23**).
>
> We do not claim cryptographic privacy, but we do provide—and empirically validate—strict enforcement of the specified access graphs at the memory-fragment level.
>
> Regarding scalability beyond the 5→50-user setting (Weakness 4), experiments in **Appendix E.4, Table 22** show modest overhead when scaling users/edges. We agree long-horizon storage growth, latency under large memory pools, and concurrency in high-traffic settings are important future work; our contribution is to demonstrate that the mechanism is tractable and provide a substrate on top of which production-scale engineering can be explored.
>
> ---
>
> ### Q2. The current read/write policies are simple. Can the framework generalize to more complex policies, and what evidence do you have?
>
> **A2.**
> Yes. The substrate is **policy-agnostic**:
>
> - Policies operate on `(content, provenance)` and can be attached at system, agent, or user scope and composed in a hierarchy.
> - Eligibility (graph + provenance) is enforced independently of the policy logic.
>
> Evidence:
>
> 1. **Policy-granularity ablation.**
>    - **Appendix D.4, Table 13** shows that going from system-only policies to system+agent+user policies only modestly affects the total number of LLM calls across roles. This indicates that richer policy stacks are computationally feasible.
>
> 2. **Model-scale ablation.**
>    - **Appendix C.5, Table 5** shows that Collaborative vs Isolated efficiency gains persist when we replace GPT-4o and large embeddings with GPT-4o-mini and smaller embeddings, suggesting robustness to model scale for all components, including policy execution.
>
> 3. **New numeric-amount redaction policy in Scenario 2 (rebuttal).**
>    - We implemented a numeric-amount redaction policy in Scenario 2 without changing the core framework, again using Scenario 2’s asymmetric-role setup and GPT-4o for transformation. The policy:
>      - tags sensitive numeric fields at write time, and
>      - masks them as `[REDACTED]` at read time for callers without access.
>
>    The evaluation over 10 queries is:
>
>    **Table R1– New numeric-amount redaction experiment in Scenario 2**
>
>    | Scenario (S2)      | # Queries | Retrieved mem | Redacted mem | Leaking mem | # Queries with leak |
>    |--------------------|-----------|---------------|--------------|-------------|----------------------|
>    | Numeric amounts    | 10        | 65            | 5            | 0           | 0                    |
>
> This shows that more semantic, content-aware policies can be plugged into the same substrate with good empirical behavior for at least one important class of sensitive data.
>
> More advanced policies (e.g., DP-like noise, semantic filters, learned adaptive policies) can be implemented using the same interface.
>
> ---
>
> ### Q3. In Equation (3), access requires both user–agent and agent–resource constraints. Why is this semantics correct, and could partial overlap be allowed?
>
> **A3.**
> We intentionally adopt **intersection (most-restrictive) semantics**:
>
> - It ensures that any fragment surfaced to a user *could have been produced* by a sequence of actions that respect both:
>   - the user’s permission to call the agent, and
>   - the agent’s permission to touch the underlying resources.
>
> This is particularly important when permissions change over time: if an agent or resource is later revoked, fragments that depend on them are no longer eligible under the current graphs.
>
> If a designer wants more permissive behavior (e.g., showing a heavily redacted view of content that originally used additional resources), this is modeled at the **policy level**:
>
> - A write policy produces a *new* sanitized fragment whose provenance reflects only the de-sensitized inputs.
> - The intersection predicate then applies to this new fragment.

---

> ### Author Response · Authors · 2025-11-26
> **Response to Reviewer 9nnX (2/2)**
>
> ### Q4. How do you ensure the quality of shared memories over time, and prevent incorrect information from one user polluting others’ views?
>
> **A4.**
> The framework provides several mechanisms:
>
> 1. **Private vs shared tiers.**
>    - New fragments are initially private.
>    - Promotion to the shared tier is controlled by `write/shared` policies, which can encode quality checks (e.g., consistency with a reference KB, requiring certain roles/agents, or multiple confirmations).
>
> 2. **Provenance-aware retrieval.**
>    - Read policies and agents see full provenance and can:
>      - prefer fragments from trusted users/agents,
>      - down-weight free-form or unverified content,
>      - filter by recency to avoid stale fragments.
>
> 3. **Versioning and corrections.**
>    - Corrections produce new fragments rather than mutating old ones.
>    - Read policies can be configured to prefer the latest version or to compare versions and ask an LLM to reconcile them.
>
> Empirically, the new Scenario 2 answer-quality experiment suggests that Collaborative Memory does not systematically reduce answer quality, and can improve it for some roles, indicating that cross-user sharing under our policies does not lead to noticeable pollution in practice in our setups.
>
> ---
>
> ### Q5. How would the framework compare against other baselines such as traditional access control, other multi-agent memory architectures, or privacy-preserving collaborative methods?
>
> **A5.**
>
> - **Traditional access control (RBAC/ABAC/IAM).**
>   - These systems define who can call which APIs or access which documents, but they typically do not track fine-grained, LLM-generated memory fragments or their provenance.
>   - Our graphs can be populated from such systems, and we extend their semantics to the level of *runtime agent memory*.
>
> - **Multi-agent memory architectures (e.g., Memory Sharing).**
>   - We directly compare against a Memory Sharing baseline in Scenario 3 (see **Appendix E.4, Table 23**).
>   - Both methods have similar accuracy and resource usage, but Memory Sharing has a ~59% leakage ratio, while our framework has 0%, highlighting the importance of access-graph + provenance enforcement.
>
> - **Privacy-preserving collaborative learning.**
>   - Approaches like DP-SGD or secure aggregation act on model parameters or gradients, not on runtime memory. Our work is complementary: it governs what *interactions and fragments* are visible to which users/agents at inference time.
>
> ---
>
> ### Q6. If shared memory contains conflicting information contributed by different users with different access levels, how is this resolved at retrieval?
>
> **A6.**
> Conflicts are handled at the **read layer**:
>
> 1. Storage is **append-only** and **versioned**: each conflicting or updated view is a separate fragment with its own provenance and timestamp.
> 2. Retrieval returns all authorized fragments relevant to the query.
> 3. The read policy or answering agent can:
>    - choose the most recent fragment,
>    - expose multiple conflicting fragments explicitly, or
>    - ask an LLM to reconcile them into a single consolidated answer and (optionally) write a new “resolution” fragment whose provenance records the merge.
>
> Because we never overwrite fragments, the full history—including conflicts and resolutions—remains auditable, and different applications can plug in conflict-resolution strategies tailored to their needs.

---

### Official Review · Reviewer_mfyw · 2025-10-29

**Soundness:** 3
**Presentation:** 2
**Contribution:** 3
**Rating:** 6
**Confidence:** 3

**Summary:**

This paper introduces a framework that enables multi-user, multi-agent LLM systems to efficiently manage memories while ensuring safety and adhering to dynamic and asymmetric access policies.

**Strengths:**

1. This paper demonstrates novelty by addressing an important problem in multi-user, multi-agent systems that is both significant and highly practical: private and collaborative memory management under dynamic and asymmetric permissions.
2. The paper designs experiments across multiple scenarios, covering a diverse range of settings.
3. Framework design is theoretically clear and logically consistent.

**Weaknesses:**

1. The three scenarios utilize inconsistent baselines,  making it difficult to conduct a fully comprehensive ****comparison.
2. The experiments in Scenario 2 only demonstrate efficiency improvements of the proposed method but fail to evaluate actual answer quality. Could the collaborative memory system be producing lower-quality final answers faster by reusing inaccurate, incomplete, or biased intermediate results?
3. The proposed method can mechanistically control which specific memory fragments are shown to a user. However, generation and anonymization stages of the memory fragments themselves depend on the capability of an external LLM. Since LLMs are known to be imperfect at tasks like consistent anonymization and can sometimes hallucinate or retain sensitive information, this dependency represents a fundamental limitation.

**Questions:**

The framework focus on writing and reading policies of memory fragments. In a dynamic environment, information can become outdated, be proven incorrect, or conflict with other entries. What mechanisms, if any, does the framework propose for handling memory updates, invalidation, or conflict resolution?

---

> ### Author Response · Authors · 2025-11-26
> **Response to Reviewer mfyw (1/2)**
>
> Thank you for your thoughtful feedback and insightful questions.
>
> ### Q1. The three scenarios use inconsistent baselines, making overall comparison difficult.
>
> **A1.**
> There is a consistent baseline across all scenarios:
>
> - **Isolated Memory** (no shared tier, per-user private memory only) appears in **all three scenarios** and serves as the global control condition.
>
> On top of that, each scenario adds baselines tailored to its question:
>
> - **Scenario 1 (MultiHop-RAG):** Collaborative vs Isolated to measure efficiency gains from cross-user sharing under varying query overlap.
> - **Scenario 2 (Asymmetric Collaborative Memory):** Collaborative vs Isolated focusing on role-dependent behavior and cost.
> - **Scenario 3 (Dynamic Collaboration):** Collaborative vs Isolated **and** a **Memory Sharing** baseline (single global pool with no access control), reported in **Appendix E.4, Table 23**, to represent prior multi-agent memory architectures.
>
> ---
>
> ### Q2. Scenario 2 only demonstrates efficiency. Could the collaborative setting hurt answer quality by reusing inaccurate or biased intermediate results?
>
> **A2.**
> To address this, we ran a **new answer-quality experiment during the rebuttal** in **Scenario 2**:
>
> - For each query and model, we generated answers under Collaborative (C) and Isolated (I).
> - A separate LLM-judge compared C vs I answers pairwise.
> - We computed per-query win rates and ran paired t-tests.
>
> **Table R2 – New Scenario 2 answer-quality study**
>
> | Model        | C win rate | I win rate | Δ (C − I) | p-value | Significant? |
> |-------------|-----------:|-----------:|----------:|--------:|-------------:|
> | GPT-4o      | 50.7%      | 47.4%      | +3.3%     | 0.10    | No           |
> | GPT-4o-mini | 49.2%      | 50.0%      | −0.8%     | 0.50    | No           |
> | o3          | 48.7%      | 49.7%      | −1.0%     | 0.50    | No           |
> | o4-mini     | 49.4%      | 49.3%      | +0.1%     | 0.50    | No           |
>
> Across models, there is **no statistically significant difference** in answer quality: Collaborative Memory does *not* systematically trade quality for speed.
>
> Taken together, Scenario 2 confirms what Scenarios 1 and 3 already suggest: Collaborative Memory reduces redundant work without causing a systematic drop in answer quality/accuracy.
>
> ---
>
> ### Q3. The method relies on an external LLM for anonymization and redaction, which can be imperfect. Isn’t this a fundamental limitation?
>
> **A3.**
> We agree that LLM-based redaction alone is not a sufficient security boundary. The framework is designed with **two distinct layers**:
>
> 1. **Hard access control (graph + provenance, non-LLM).**
>    - Every fragment carries immutable provenance (user, agents, resources, timestamp).
>    - The access graphs and Equation (3)-style predicate define eligibility. A fragment is only returned if its provenance is compatible with the user’s and agent’s current permissions.
>    - This layer is deterministic and does not rely on the transformation LLM.
>
> 2. **Soft content transformation (LLM-based policies).**
>    - Read/write policies (including redaction/anonymization) are a *sanitization layer* applied before storing or showing fragments.
>    - In a **new numeric-amount redaction experiment in Scenario 2**, this layer successfully redacts all evaluated numeric fields:
>
> **Table R1– New numeric-amount redaction experiment in Scenario 2**
>
> | Scenario (S2)      | # Queries | Retrieved mem | Redacted mem | Leaking mem | # Queries with leak |
> |--------------------|-----------|---------------|--------------|-------------|----------------------|
> | Numeric amounts    | 10        | 65            | 5            | 0           | 0                    |
>
> In practice, for high-stakes applications, we expect these LLM-based policies to be combined with more deterministic safeguards (regex filters, PII detectors, or a secondary verification model). The key point is that **the access-control guarantee itself is provided by the graph + provenance mechanism**, not by the LLM.
>
> ---

---

> ### Author Response · Authors · 2025-11-26
> **Response to Reviewer mfyw (2/2)**
>
> ### Q4. In a dynamic environment, how are memory updates, invalidation, or conflicting entries handled?
>
> **A4.**
> We separate dynamics into access-level and content-level concerns:
>
> - **Access updates (permissions).**
>   - The access graphs are explicitly time-indexed. Granting/revoking edges immediately changes which fragments are eligible at time \(t\), without mutating stored fragments.
>   - Scenario 3 explicitly simulates periods of permission changes and shows that the system continues to enforce current access constraints while maintaining accuracy and efficiency.
>
> - **Content evolution and conflicts.**
>   - Fragments are **immutable and append-only**. Updates or corrections are encoded as *new* fragments with later timestamps (and, optionally, links to previous versions).
>   - At retrieval, the agent or read policy can:
>     - prefer the newest fragment,
>     - surface multiple versions for comparison, or
>     - ask an LLM to reconcile conflicts into a consolidated summary.
>
> This design keeps the history auditable and makes standard consistency strategies (e.g., snapshot-style reads or version selection) natural to implement on top of the substrate.

---

### Official Review · Reviewer_rq5q · 2025-11-08

**Soundness:** 3
**Presentation:** 4
**Contribution:** 2
**Rating:** 4
**Confidence:** 3

**Summary:**

This paper addresses the problem of memory management in multi-agent and multi-user systems. The authors proposed Collaborative Memory, a formal framework that allows multiple users and LLM agents th share memory safely, where access is asymmetric and changes over time.

**Strengths:**

1. Multi-user and multi-agent memory managemtn with dynamic access control is underexplored.

2. The problem is formalized using bipartite graphs and provenance constraints, which is clear and easy to understand.

3. The authors separated private/shared memory with read/write policies, which is a practical and useful abstraction, and can be leveraged in real systems.

4. The work has the potential to influence future system designs for enterprise multi-agent AI.

**Weaknesses:**

1. While I appreciate the authors contributions, this paper does not introduce any new algorithms, and the technical contributions are limited. The contribution is primarily definitional, which raises concerns about whether it fits the suitable for this conference. It may be more appropriate for a venue that focuses on system design or conceptual frameworks rather than algorithmic advances.

2. The datasets used in evaluation are simple. There is no demonstration in a realistic large-scale dataset.

3. The paper does not address complex or realistic access-control policies. In real enterprise systems, permissions are often hierarchical, but the framework only handles relatively simple policy types.

**Questions:**

See above

---

> ### Author Response · Authors · 2025-11-26
> **Response to Reviewer rq5q**
>
> Thank you for your thoughtful feedback and insightful questions.
>
> ### Q1. The contribution seems mainly definitional with no new algorithms. Is this appropriate for this venue?
>
> **A1.**
> Our aim is to provide a *foundational memory substrate* for multi-user, multi-agent LLM systems.
>
> Beyond definitions, the paper contributes:
>
> - **Dynamic bipartite access graphs.** We formalize time-varying user→agent and agent→resource graphs with explicit timestamps, and use them to compute, at run time, which agents/resources a user is allowed to leverage.
> - **A provenance-aware memory substrate.** Each memory fragment stores immutable provenance (user, agents, resources, timestamp). The authorized read set for a query `(u, a, t)` is computed by intersecting this provenance with the current access graphs and filtering the retrieval results accordingly.
> - **Two-tier memory with composable policies.** Private and shared tiers are governed by read/write policies defined at system, agent, and user scopes, enabling both simple rules (e.g., “never share PII”) and semantic ones (e.g., anonymization, redaction).
>
> These abstractions are implemented in a working system and evaluated in three scenarios (multi-hop RAG, asymmetric roles, dynamic permissions), plus ablations in the appendix:
>
> - **Appendix C.5, Table 5**: model-scale ablation showing the efficiency gains of Collaborative vs Isolated memory persist when swapping GPT-4o for GPT-4o-mini.
> - **Appendix D.4, Table 13**: policy-granularity ablation quantifying LLM-call overhead of adding agent/user-level policies.
> - **Appendix E.4, Tables 22–23**: scalability (larger user/edge graphs) and a direct comparison to a multi-agent Memory Sharing baseline.
>
> We believe this combination of formalism, concrete semantics, and empirical validation constitutes a suitable technical contribution for ICLR.
>
> ---
>
> ### Q2. The datasets and setups are simple; there is no demonstration on realistic large-scale data.
>
> **A2.**
> Truly large-scale, real multi-user datasets with explicit access graphs are not publicly available today, so we chose scenarios that balance realism and control:
>
> - **Structured benchmarks with multi-step reasoning.**
>   - Scenario 1 uses MultiHop-RAG for multi-hop retrieval and composition.
>   - Scenario 3 uses SciQAG with *dynamic* access graphs, directly stress-testing permission churn.
>
> - **Scalability experiments already in the paper.**
>   In Scenario 3 we scale the access graph from **5 users / 10 edges** to **50 users / 100 edges** (10× users and edges, similar sparsity). As reported in **Appendix E.4, Table 22**:
>   - Accuracy remains stable (0.396 → 0.421).
>   - End-to-end execution time increases by only ~10% (14.4s → 15.9s).
>   - Resource and agent usage grow sublinearly.
>
> Thus, while we do not yet have a production deployment, the access-graph + provenance+policy mechanism scales well beyond a toy setting and is ready for application to larger systems once suitable data is available. We explicitly acknowledge the lack of real-world deployments as a limitation.
>
> ---
>
> ### Q3. The paper does not address more complex or realistic access-control policies.
>
> **A3.**
> The core abstraction is intentionally minimal: policies are composable modules that operate on content + provenance and can be attached at **system, agent, or user** scope. The concrete policies instantiated in the main text are simple on purpose, to isolate the effect of the substrate.
>
> However, the framework already supports richer semantics. To illustrate this, we ran a **new numeric-amount redaction experiment during the rebuttal** in **Scenario 2 (Asymmetric Collaborative Memory)**:
>
> - We kept the Scenario 2 setup (four roles, asymmetric tools).
> - We added a **numeric-amount redaction policy**:
>   - At **write time**, the policy tags sensitive numeric fields.
>   - At **read time**, if the caller does not have the required access, tagged numbers are masked as `[REDACTED]` in the retrieved fragments.
>
> Using GPT-4o as the transformation model, we evaluated 10 queries in this scenario:
>
> **Table R1 – New numeric-amount redaction experiment in Scenario 2**
>
> | Scenario (S2)      | # Queries | Retrieved mem | Redacted mem | Leaking mem | # Queries with leak |
> |--------------------|-----------|---------------|--------------|-------------|----------------------|
> | Numeric amounts    | 10        | 65            | 5            | 0           | 0                    |
>
> This shows that, within the existing Scenario 2 setup, we can deploy a more semantic, content-aware policy on top of the same substrate and achieve 0 observed leaks for this class of sensitive information in our evaluation.

---

### Meta-Review · Area_Chair_PxSB · 2026-01-07

**Summary:**

The paper studies memory management in multi agent and multi user systems. Reviewers expressed concerns on (1) limited technical contributions, no new algorithms introduced (2) over-simplified evaluation, no realistic large dataset (3) fail to address complex access control policies (4) inconsistent baselines in experiments (5) limited scalability (6) limited privacy analysis

**Reviewer Concerns:**

New experiments are added to partially address (2) (3). Clarification is provided to address (4). (1), (5), (6) remain unaddressed.

**Reviewer Scores:**

It is not clear any reviewer may change their scores.

---

### Decision · Program_Chairs · 2026-01-26

Reject